# LONG COT IN-CONTEXT LEARNING CAN EMPOWER PRE-TRAINED LLMS

## ABSTRACT

Recent advances in Large Reasoning Models (LRMs) highlight the importance of long chain-of-thought (CoT) reasoning for complex tasks. However, most existing methods rely post-training that tunes the model parameters, obscuring whether pre-trained models intrinsically possess such capabilities. We propose in-context learning (ICL) with long CoT demonstrations as a tuning-free approach to investigate this. Across Qwen 2.5 (7B, 32B) and DeepSeek V3 models on mathematical reasoning tasks, we demonstrate that ICL empowers base models to exhibit sophisticated long CoT behaviors like reflection and verification. Furthermore, it delivers performance gains (pass@1–pass@K) over direct generation, supporting the conjecture that base models possess inherent reasoning capabilities, but not fully leveraged by direct prompting. Furthermore, our in-depth analysis reveals that long CoT ICL not only improves accuracy on easy problems but also enables models to solve previously intractable medium problems. Finally, we validate that tasks benefit from long CoT ICL when problem-relevant demonstrations are provided. For instance, given problem-relevant demonstrations, the performance of DeepSeek V3 on AIME25 improves by 6.5%. We hope this work could advance the understanding of the mechanisms and intrinsic abilities of long CoT reasoning.

## 1 INTRODUCTION

Recent advances in large language models (LLMs) have introduced a new class of Large Reasoning Models, such as OpenAI o1 (OpenAI, 2024), DeepSeek-R1 (Guo et al., 2025), and Qwen3 (Yang et al., 2025). These models generate explicit and structured reasoning before producing final answers, a process commonly referred to as long chain-of-thought (long CoT) inference (Chen et al., 2025; Li, 2025). This paradigm integrates CoT with iterative exploration and reflection, which can significantly enhance a model's ability to solve complex reasoning tasks (Wang et al., 2025).

Despite these impressive results, eliciting the long CoT reasoning capabilities of LLMs remains an open challenge. Most existing methods involve post-training to tune model parameters (Guo et al., 2025; Team et al., 2025; Yang et al., 2025), which obscures whether pre-trained models intrinsically possess such capabilities. While Yeo et al. (2025a) demonstrated that long CoT data patterns exist in pre-training corpora like OpenWebMath (Paster et al., 2023), recent research indicates that zero-shot prompting methods struggle to elicit long CoT reasoning from pre-trained models, as they are constrained by the base model's inherent solution space (Yeo et al., 2025a; Yue et al., 2025). The primary challenge lies in guiding models to explore beyond this intrinsic solution space.

Inspired by previous research on in-context learning (ICL) (Brown et al., 2020; Agarwal et al., 2024) as a tuning-free paradigm for steering model behaviors, we investigate whether ICL with long-CoT demonstrations can empower base models to exhibit long CoT behaviors. This task presents a significant challenge, as unlike the often easy and short demonstrations used in prior ICL studies, long-CoT demonstrations are considerably more complex, demanding both sophisticated understanding of the input and the ability to generate elaborate outputs.

In this paper, we first demonstrate that long-CoT in-context learning (ICL) prompting can induce pre-trained models to exhibit long CoT patterns on mathematical tasks, as illustrated in Figure 1. Specifically, we quantify long CoT patterns in models' outputs and experiment with the Qwen2.5-7B base model (Yang et al., 2024) using demonstrations generated by DeepSeek-R1. We find that the proportion of deep reasoning behaviors, such as reflection and verification, significantly increases

compared to direct generation. This observation suggests that pre-trained models can be prompted to exhibit long CoT patterns without parameter fine-tuning, essentially activating a "reasoning style".

Beyond this "style activation", we further examine whether this elicitation translates into improved task performance. To this end, we study a broader set of pre-trained models, including the Qwen2.5 family (7B, 32B) (Yang et al., 2024) and DeepSeek V3 (Liu et al., 2024). Testing on diverse mathematical reasoning tasks, we consistently observe performance improvements, indicating that the models discover better solutions through the induced long CoT reasoning.

What accounts for these observed performance improvements? We conduct an in-depth analysis of Qwen2.5-32B's performance on the AIME25 benchmark across questions of varying difficulty levels (easy, medium, and hard). This analysis reveals that long-CoT ICL not only enhances accuracy on easy problems but also enables the model to tackle previously intractable medium problems. And then, we investigate this question, hypothesizing that they stem from two potential factors: the emergence of long CoT behaviors (i.e., "style activation") or genuine gains in the model's intrinsic reasoning ability. As detailed in Section 4.1.3, our findings indicate that the observed performance boost is primarily attributable to the former, while the model's fundamental reasoning ability shows no significant enhancement.

Finally, to further optimize the performance of long-CoT ICL, we study various factors that may affect its efficacy, such as the source and number of demonstrations. In particular, we validate that tasks particularly benefit from long-CoT ICL when problem-relevant demonstrations are provided. For instance, DeepSeek V3's performance on AIME25 improves by 6.5% under such conditions.

In summary, our contributions are:

- We propose in-context learning (ICL) with long-CoT demonstrations to explore whether and how pre-trained models' long CoT reasoning capabilities can be elicited.

- We conduct comprehensive experiments on mathematical reasoning tasks, showing that ICL with long CoT improves accuracy by eliciting long CoT behaviors.

- We conduct in-depth analysis, which reveals that long CoT ICL not only enhances accuracy on easy problems but also enables the model to tackle previously intractable medium problems.

- We validate that long CoT ICL yields greater performance gains when problem-relevant demonstrations are provided, compared to random selection.

## 2 Preliminaries

### 2.1 Long CoT Prompting

Chain-of-Thought (CoT), first introduced by Wei et al. (2022), is a technique that guides large language models LLMs to explicitly produce intermediate reasoning steps before delivering a final answer. OpenAI has validated that *test-time scaling* can substantially improve performance on complex tasks by allocating more compute at inference (OpenAI, 2024). A key phenomenon is that the model's reasoning becomes increasingly fine-grained, often accompanied by behaviors such as reflection and the exploration of alternative solutions, collectively referred to as long CoT. Following Chen et al. (2025), we characterize long CoT along three key mechanisms: (1) **Deep reasoning**, by extending the allowable reasoning length from a short CoT boundary ($B_s$) to a long CoT boundary ($B_\ell$), where $B_\ell \gg B_s$. (2) **extensive exploration**, by encouraging branching out to extensively explore uncertain or unknown logical paths; and (3) **feasible reflection**, by allowing iterative revisitation and refinement of earlier steps. These mechanisms jointly increase a model's ability to handle complex tasks. With the emergence of o1, there is growing interest in generating long CoT reasoning. While most existing approaches rely on post-training to produce long CoT reasoning paths (Ye et al., 2025; Muennighoff et al., 2025; Guo et al., 2025; Team et al., 2025), a few studies have attempted to generate long reasoning traces without fine-tuning, for example by appending control words such as `wait` (Yeo et al., 2025b; Shen et al., 2025a).

## 2.2 IN-CONTEXT LEARNING

Let $M$ be a pretrained large language model. Given an input prompt $x_{prompt}$ structured as a sequence of $k$ input-output demonstrations $\mathcal{S} = \{(x_1, y_1), \ldots, (x_k, y_k)\}$ followed by a new query $x^*$, In-Context Learning (ICL) refers to the ability of $M$ to perform the task exemplified by $\mathcal{S}$ on $x^*$ without updating its parameters. Formally, the model predicts the output $\hat{y}$ for the query $x^*$ by maximizing the conditional probability:

$$\hat{y} = \arg\max_{y_j \in \mathcal{Y}} P_M(y_j \mid x^*, \mathcal{S})$$

where $P_M$ denotes the probability assigned by the model $M$ to the output $y_j$ conditioned on the query $x^*$ and the in-context demonstrations $\mathcal{S}$. This entire process occurs within the model's forward pass, without any gradient updates to $M$'s parameters.

By presenting the model with relevant examples or demonstrations, this approach enables few-shot learning, reducing the need for task-specific fine-tuning. ICL can be effectively combined with CoT. For example, providing step-by-step reasoning examples in the prompt can help LLMs generalize to unseen tasks, making in-context learning a powerful tool for improving reasoning capabilities. By providing ICL samples from specific domains, the model can better automate prompt design (Zhang et al., 2023) and actively engage in prompting (Diao et al., 2024), as well as perform tree search (Yao et al., 2023). ICL operates as a form of algorithm execution within the model's forward pass, where architectural features like "induction heads" (Olsson et al., 2022) infer and apply task structure from contextual examples. In this work, we investigate the impact of long CoT ICL on the emergence of long CoT reasoning patterns.

# 3 LONG CoT IN-CONTEXT LEARNING EMPOWERS PRE-TRAINED LLMS

## 3.1 SETTINGS

**Demonstration** We focus on mathematical reasoning tasks. To construct demonstrations, we randomly draw questions from DeepScaler (Luo et al., 2025) dataset and pair them with responses generated by DeepSeek R1, whose outputs exhibit long CoT reasoning patterns. We select only cases in which the LLM provides correct answers, ensuring that the demonstrations reflect both accuracy and extended reasoning behavior. For comparison, we also collect short CoT demonstrations from OpenAI O1 (OpenAI, 2024) for short CoT ICL.

**Models and Benchmarks** To ensure the robustness of conclusions, we experiment with multiple LLM families, primarily Qwen2.5 (7B/32B base model) (Yang et al., 2024) and DeepSeek V3 (Liu et al., 2024). We evaluate our approach on mainstream datasets, including AIME25, MATH500 (Hendrycks et al., 2021), AMC23, and MinervaMath (Lewkowycz et al., 2022). For AIME25, we use the full test set, while for MATH500 and MinervaMath, we randomly select subsets of 50 problems each as the test set.

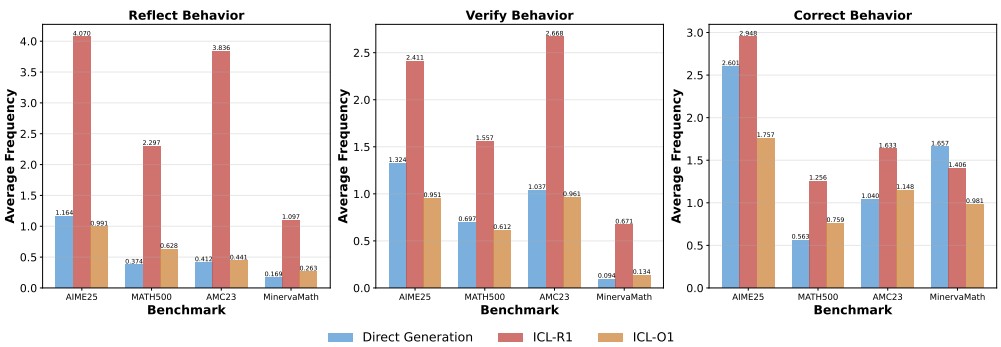

Figure 1: Long CoT Behavior on Qwen2.5-7B

**Evaluation Protocol** For the sampling procedure across all models, we use commonly adopted parameters: a temperature of 0.6, a top-p value of 0.95, and a maximum response length of 16,384 tokens. For ICL generation, we employ four demonstrations randomly selected from the DeepScaler dataset for small models and two for DeepSeek V3.

**Pass@K Metric** To evaluate the model's reasoning ability, we employ the pass@K metric with rule-based rewards across all tasks. Given a problem, we sample K responses, each of which is scored using the rule-based reward: correct answers are assigned a value of 1, while incorrect answers are assigned 0. Over the entire dataset, we compute the average pass@K across all questions, which reflects the proportion of problems that can be correctly solved within K trials. In practice, we adopt the unbiased estimator of pass@K, as described in Appendix A.1.1.

**Baselines** We mainly consider two baselines: **Direct Generation (DG)**: the model takes the problem as input and directly generates a CoT; and **ICL-O1**, in which the model is prompted with short CoT demonstrations derived from OpenAI O1.

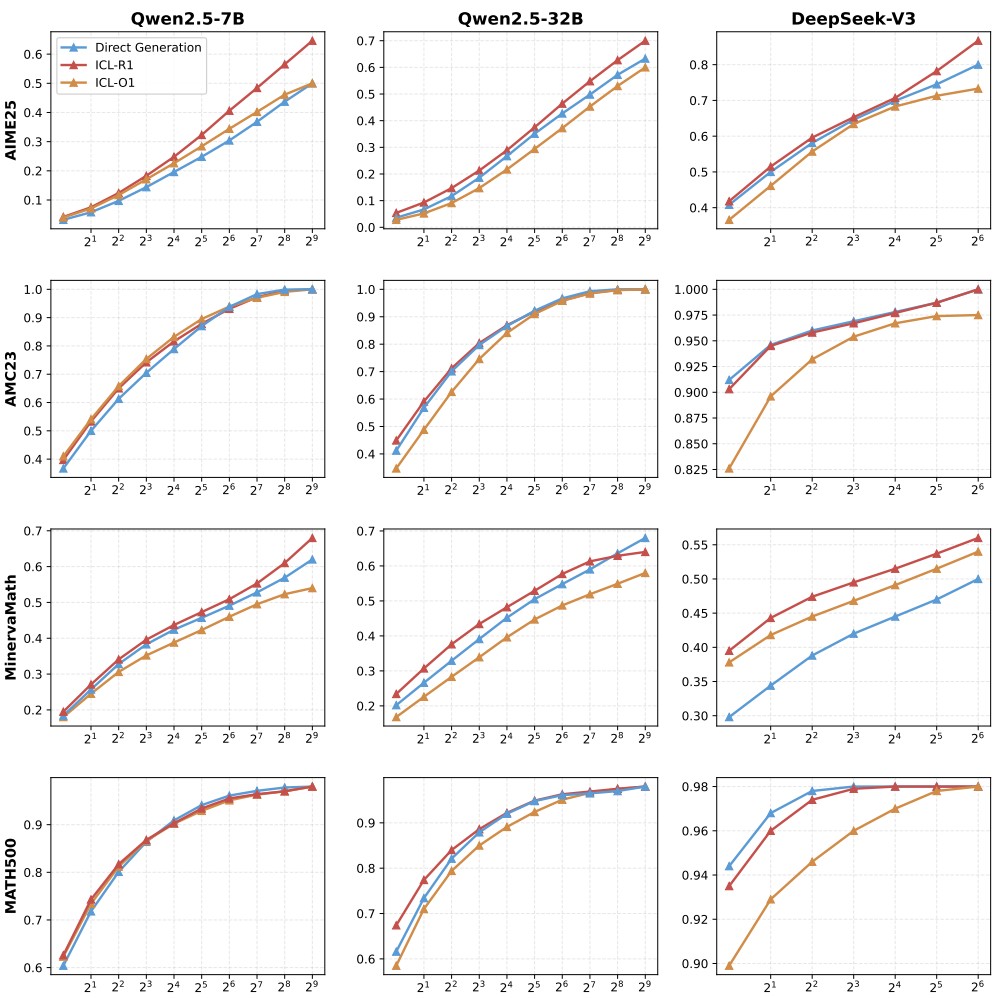

Figure 2: The performance comparison between ICL with long CoT demonstration, direct generation and ICL with short CoT demonstration.

## 3.2 LONG COT ICL CAN ACTIVATE LONG COT PATTERN

To verify the impact of long CoT in-context learning on pre-trained models, we first present long CoT ICL prompting named **ICL-R1** can enable pre-trained models to exhibit long CoT patterns.

Specifically, we utilize the Qwen2.5-7B base model with demonstrations generated by DeepSeek-R1 and quantify its long CoT behavior using the reasoning behavior ratio metric, described in Appendix A.1.2. As shown in the Figure1, across all benchmarks, the model's outputs under long CoT ICL exhibit a higher frequency of reflection, verification, and correction behaviors. In particular, the frequency of reflection is approximately $4\times$ that observed in outputs produced by direct generation. The results reveal that long CoT behaviors, such as long CoT behaviors like reflection and verification, emerge with higher frequency compared to direct generation. These findings suggest that pre-trained models, even without post-training on long CoT datasets, can exhibit sophisticated long CoT behaviors simply by providing long CoT demonstrations.

### 3.3 Long CoT ICL can improve the performance of pre-trained LLMs

Building on this insight, we further investigate the impact of long CoT ICL on model performance. Specifically, we extend our study to a broader set of pre-trained models, including the Qwen2.5 family (7B and 32B) and DeepSeek V3, and evaluate them across diverse benchmarks. For each problem, we randomly select demonstrations for pre-trained models. For comparison, we employ short CoT ICL and direct generation as baseline conditions. The pass@K results are shown in Figure 2. We also report pass@1 performance in Table 1. Our experiments show that long CoT ICL improves the model's pass@K performance. In particular, substantial gains in pass@1 are observed on tasks such as AMC23, Math500, and MinervaMath, whereas the improvement on AIME25 is relatively limited due to its higher difficulty. We observe that the improvements of DeepSeek V3 on AMC23 and Math500 are limited. We hypothesize that this is because the model is already capable of solving many of these problems directly, and the addition of ICL demonstrations may instead introduce noise, thereby reducing effectiveness. Moreover, for the Qwen family of models, performance gains become increasingly pronounced as model size grows, a trend we attribute to the enhanced ability of larger models to follow ICL demonstrations. These findings provide evidence that long CoT ICL promotes performance beyond the base model.

Table 1: Pass@1 performance across benchmarks. Best performance are bold.

| Model | Method | AIME25 | AMC23 | MATH500 | MinervaMath | Average |
|---|---|---|---|---|---|---|
| Qwen2.5-7B | DG | 3.2 | 36.7 | 60.4 | 18.4 | 29.7 |
| | ICL-O1 | 4.0 | **41.0** | 62.3 | 18.0 | 31.3 |
| | ICL-R1 | **4.2** | 39.7 | **62.6** | **19.7** | **31.6** |
| Qwen2.5-32B | DG | 3.6 | 41.2 | 61.6 | 20.2 | 31.7 |
| | ICL-O1 | 2.8 | 34.7 | 58.5 | 16.8 | 28.2 |
| | ICL-R1 | **5.4** | **44.9** | **67.4** | **23.4** | **35.3** |
| DeepSeek V3 | DG | 40.8 | **91.2** | **94.4** | 29.8 | 64.1 |
| | ICL-O1 | 36.6 | 82.6 | 89.9 | 37.8 | 61.7 |
| | ICL-R1 | **41.8** | 90.3 | 93.5 | **39.5** | **66.3** |

## 4 Deep Analysis

In this section, we further analyze the causes of the performance improvements and investigate why some problems remain unsolved. We then explore the relationship between long CoT ICL and fine-tuning, and finally explore the performance if relevant demonstrations were provided.

### 4.1 Why long CoT can or can not improve reasoning performance

To investigate the source of performance improvements, we follow Sun et al. (2025) and analyze per-problem performance of Qwen2.5-32B on AIME25. Specifically, we categorize problems into three difficulty levels based on pass@1 accuracy under long CoT ICL. Easy-level questions are those typically solvable by long CoT ICL, for which pass@128 performance approaches 100%. Hard questions are those that cannot be solved by long CoT ICL, and the remaining problems are classified as medium-level. To facilitate comparison, we report both pass@8 and pass@128 scores across the different categories. From Figure 3, we observe that for easy questions, long CoT ICL

substantially improves accuracy, as reflected in the pass@8 results. For medium questions, long CoT ICL also enhances performance, as reflected in the pass@128 results. Specially, for Problems 23 and 28, the model is able to solve problems that were previously intractable.These results suggest that for problems the model is already capable of solving, long CoT ICL enhances overall performance. Next, we manually check the questions to identify how the model's behavior varies compared with direct generation.

### 4.1.1 WHY CAN LONG CoT ICL IMPROVE REASONING PERFORMANCE?

**Refining Algebraic Derivation and Problem-Solving Process.**    For mathematical problems, language models solve mathematical problems primarily through step-by-step variable solving and formula application. When using the Long CoT ICL, the model not only follows the CoT reasoning style in its overall solution process but also consistently carries it out in the concrete solving steps. For example, in the answer to Problem 7 in Appendix D.1, the base model directly provides the equation of the perpendicular bisector of points $4 + k$ and $3i + k$ without showing the intermediate calculation process. In contrast, the ICL method provides detailed calculations for finding the slope $\frac{4}{3}$ of the perpendicular bisector and the process of passing through point $2 + \frac{3i}{2} + k$, thereby reducing errors in intermediate steps.

**Reducing Hallucinations in Generated Code.**    The model tends to attempt problem-solving using Python code. However, without the ability to access external tools, it often produces incorrect answers. By long CoT ICL, the model solves the problem step by step, carefully enumerating possibilities, and ultimately reaching the correct solution. For example, in Problem 8 (see Appendix D.1 for details), Qwen2.5-32B exhibits severe hallucinations under direct answering, producing a segment of Python code and omitting critical intermediate reasoning steps. In contrast, with long CoT ICL, when calculating the intersection points between the parabola $y = x^2 - 4$ and its rotated image, the model does not directly provide code to obtain the intersections as the base model does, but instead adopts a reasoning style similar to R1 and solves the equations step by step, ultimately arriving at the correct result.

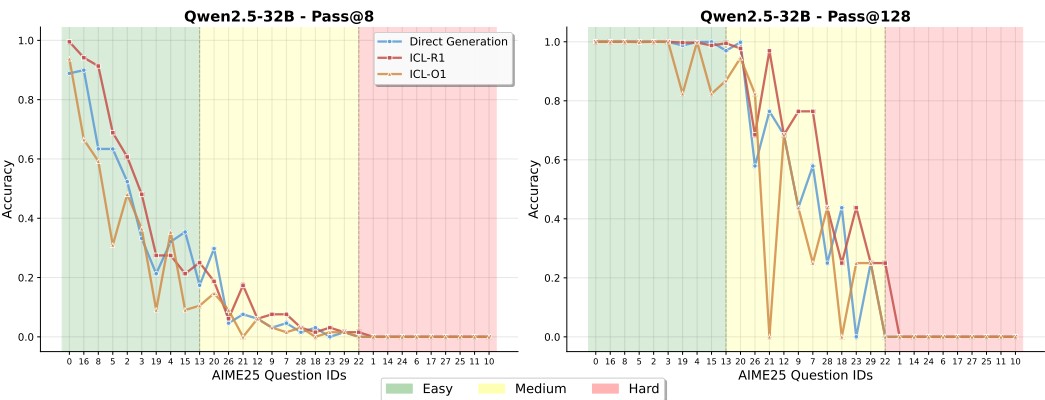

Figure 3: Pass@8 and Pass@128 on AIME25 with Qwen2.5-32B

### 4.1.2 WHY CANNOT LONG CoT ICL IMPROVE REASONING PERFORMANCE?

From Figure 3, we observe that for some problems, performance decreases under long CoT ICL, while for others, neither long CoT ICL nor direct generation is able to solve them. In this section, we analyze these cases in detail.

**Higher Rate of Geometric Misjudgments.**    The long CoT demonstrations can cause misunderstandings in the order of points and the logical relationships of edges in geometry problems. For example, given three points A, B, and C on a straight line from left to right, the LLM may incorrectly infer during reasoning that AB = AC + CB, leading to wrong results. A detailed example of the ICL results shows that ICL leads to a higher rate of geometric misjudgments in the problem of

Appendix D.1. In problems 26 and 12, even though logical reasoning plays a larger role, the geometric errors introduced by ICL offset the advantages brought by logical reasoning, resulting in overall performance comparable to the base model. This suggests that randomly selected demonstrations may introduce noise into the generation process, leading to misleading responses from the model.

**Incomplete Base Model Knowledge and Problem-Solving Skills.** For hard problems, we observe that the primary source of failure lies not in the absence of relevant knowledge but in the inability to apply it effectively. For instance, in Problem 27 (see Appendix D.1), a process of taking the modulus of a large number requires the use of the Chinese Remainder Theorem. Qwen2.5-32B fails to invoke this theorem and thus produces incorrect reasoning. However, when explicitly asked "Do you know the Chinese Remainder Theorem?", the model can state the theorem correctly, indicating that the knowledge itself is present in its parameters. The difficulty therefore stems from the failure to activate and apply the related knowledge. This suggests that the challenge lies not in knowledge acquisition, but in retrieving and applying knowledge already stored within the LLM.

### 4.1.3 LONG COT BEHAVIORS IMPROVE THE PERFORMANCE OF PRE-TRAINED LLMS

The improvements of performance may arise from two factors: the emergence of long CoT behaviors and genuine gains in reasoning ability. It remains uncertain whether the model's reasoning ability has been improved. To evaluate reasoning ability, we examine three aspects: problem comprehension, adherence to a valid problem-solving strategy, and the number of correctly executed intermediate steps. We leverage an oracle model, OpenAI GPT-5 (OpenAI, 2025), to comprehensively evaluate the reasoning ability of LLMs. Specifically, the assessment comprises three aspects: Problem Understanding (0.1 points), Valid Problem-Solving Strategy (0.1 points), and Step Execution (0.8 points), which measures how many of the key reasoning steps the model executes correctly. When evaluating Step Execution scores, we provide the key steps of the solution, which are extracted from the correct answers generated by GPT-5. It is worth noting that GPT-5 failed to answer five questions correctly. The detailed process is shown in Appendix B.1.

We randomly selected 4 problems from the AIME25 dataset using Qwen2.5-32B for illustration, with additional results provided in Figure 9. The results are shown in Figure 4. For these four problems, the model's reasoning ability shows little improvement under long CoT ICL compared with direct generation, suggesting that long CoT ICL provides limited gains in reasoning ability. This may be because the randomly selected demonstrations contain only long CoT pattern information without problem-relevant knowledge. Based on the results, we conclude that the observed performance gains primarily stem from the long CoT patterns. Furthermore, since reasoning and pre-trained models differ in both pattern and reasoning ability, long CoT ICL can help mitigate differences arising from patterns, allowing for a fairer comparison of reasoning capabilities.

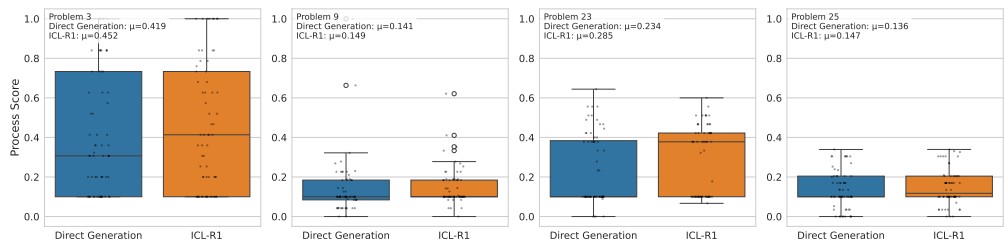

Figure 4: Score distributions of reasoning ability between Long CoT ICL and Direct generation

### 4.2 THE RELATIONSHIP WITH LONG COT FINE-TUNING

In this section, we investigate how long CoT ICL differs from post-training especially SFT in eliciting long CoT reasoning. Specifically, we compare long CoT ICL with two types of post-trained models: DeepSeek-R1-Distill-Qwen-7B (Guo et al., 2025) and S1K-7B (Muennighoff et al., 2025).The S1K-7B model is trained from the Qwen2.5-7B base model using the same configuration as in Muennighoff et al. (2025). On the AIME25 benchmark, we observe that models exhibit substantially higher pass@1 performance compared to long CoT ICL. To investigate why their capability falls

short of fine-tuning, we analyze reasoning quality across different models as shown in Figure 5. Relative to the performance from long CoT ICL prompting, the S1K-7B model achieves only certain improvements in reasoning quality, whereas the DeepSeek-R1-Distill-Qwen-7B exhibits substantially greater enhancements. We find that the improvements from Long CoT ICL primarily stem from the emergence of Long CoT patterns, whereas the gains from S1K arise from both the pattern and enhanced reasoning ability. With larger-scale fine-tuning, the model's reasoning ability could be further improved.

Table 2: Pass@1 performance on AIME25 with different numbers of shots. Best performance are bold.

| Model | 1-shot | 2-shot | 4-shot | 6-shot | 8-shot |
|---|---|---|---|---|---|
| Qwen2.5-32B | 4.0 | 3.5 | **5.4** | 4.5 | 5.0 |
| DeepSeek V3 | 39.5 | **41.8** | 37.2 | 32.9 | 33.3 |

## 4.3 ABLATION STUDY

In this section, we analyze factors influencing model reasoning ability, with a particular focus on the impact of different numbers of shots and the sources of demonstrations. By performing ablation studies on the number of shots, we identify the optimal shot count. We further investigate whether providing additional relevant knowledge can improve model performance and examine the upper bound of such improvements.

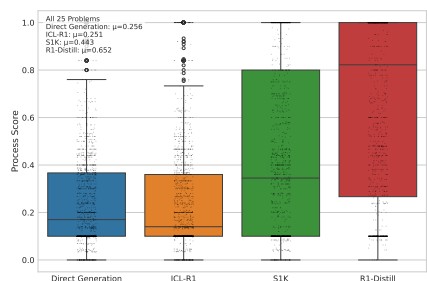

Figure 5: Average score distributions of reasoning ability for different methods

**Ablation for demonstration numbers** We conduct ablation studies to evaluate the robustness of our method under different numbers of shots, using Qwen-32B and DeepSeek V3. The results are shown in Table 2. We observe that as the number of shots increases, model performance initially improves but then declines. We speculate that, under the few-shot setting, the limited and randomly selected examples can introduce noise, which may offset the benefits of additional shots.

**ICL with Relevant long CoT Demonstration** As discussed in Section 4.1.2, the model fails to answer correctly because it cannot effectively retrieve and utilize the knowledge stored in the LLM. This raises the question of how well the model can perform when given problem-relevant demonstrations. To examine this, we used semantic matching to identify relevant problems from DeepScaler. Nonetheless, the retrieved examples were not truly aligned with the target problems. We therefore tried distilling problem-relevant demonstrations from LLMs. The detailed construction process is provided in Appendix C.2. We conducted experiments with Qwen2.5 and DeepSeek V3 on the AIME25 benchmark. The results indicate that providing problem-relevant demonstrations yields a substantial improvement over long CoT ICL with randomly selected demonstrations. The performance gain becomes more pronounced as the model's capability increases, indicating that the model can further leverage its potential when relevant knowledge is provided. However, the improvement is still less significant than what is typically achieved through fine-tuning. We speculate that this is because the model cannot fully exploit the knowledge in ICL. Namely, it has not yet learned to actively utilize the knowledge in the demonstrations when solving problems.

## 5 RELATED WORK

### 5.1 LONG CHAIN-OF-THOUGHT REASONING

Recent studies demonstrated that enabling LLMs to generate long CoT sequences during test-time inference significantly enhanced reasoning accuracy (Brown et al., 2024; Snell et al., 2024). Current researches focus on training models with long CoT reasoning through fine-tuning. By constructing and leveraging long CoT demonstrations, fine-tuning enables LLMs to generate long CoT reasoning paths that exhibit deep reasoning, extensive exploration and reflection. Specifically, Deepseek

R1 (Guo et al., 2025), extending Deepseek R1 Zero, warmup with high-quality cold-start data and utilize pure RL to achieve reasoning performance on par with OpenAI's O1 models (OpenAI, 2024). Kimi K1.5 (Team et al., 2025) utilized a high-quality long CoT dataset, employing SFT as a warmup phase that improved the generation of logically coherent and detailed responses. LIMO (Ye et al., 2025) and s1 (Muennighoff et al., 2025) challenged the necessity of large sample sizes, demonstrating that minimal sample sets successfully activated reasoning capabilities in foundational LLMs. Satori (Shen et al., 2025b) introduced a critic model for constructing multi-step demonstrations with reflection mechanisms, facilitating enhanced multi-step reasoning capabilities in trained models. This work does not rely on additional training. Instead, it activates long chain-of-thought reasoning patterns through prompting, offering greater flexibility across tasks.

## 5.2 IN-CONTEXT LEARNING

In-Context Learning (ICL) leverages contextual examples, which contains the formulation of target math reasoning abilities, provided within the prompt to guide LLMs to learn to solve new problems. By formalizing algorithmic processes as skills and incorporating them as examples (Zhou et al., 2022), the model is taught how to leverage algorithms for reasoning rather than simply engaging in imitation learning. In addition to formalizing reasoning examples as algorithms to solve problems, Jie & Lu (2023) suggests that the reasoning process can be represented through code, which effectively enables the acquisition of multi-step reasoning capabilities. Zhang et al. (2024) finds that learning from incorrect examples can also lead to improvements. Additionally, ICL can be effectively combined with CoT. For example, providing step-by-step reasoning examples in the prompt can help LLMs generalize to unseen tasks, making in-context learning a powerful tool for improving reasoning capabilities. By providing ICL samples from specific domains, the model can better automate prompt design (Zhang et al., 2023) and actively engage in prompting (Diao et al., 2024), as well as perform tree search (Yao et al., 2023). However, since there is no gradient propagation for learning, currently ICL still faces significant challenges in generalization and corresponding interpretability (Opedal et al., 2024).

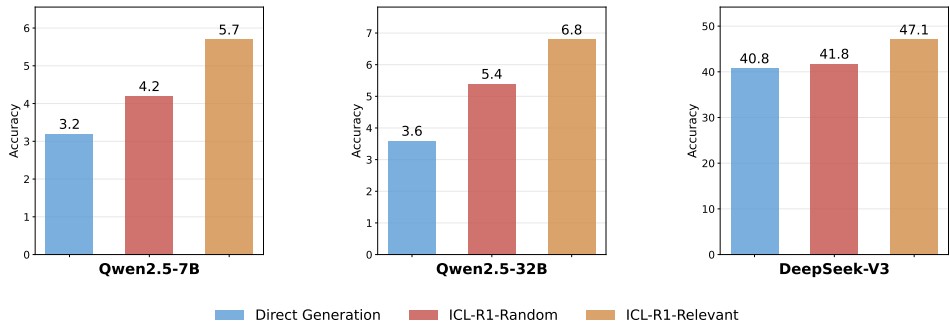

Figure 6: ICL performance with different long CoT demonstration.

## 6 CONCLUSION

In this work, we investigated the role of long Chain-of-Thought (CoT) In-Context Learning (ICL) as a tuning-free method for enhancing reasoning capabilities in pre-trained, pre-trained language models. Our experiments on Qwen2.5 and DeepSeek V3 demonstrate that long CoT ICL can effectively empower these base models to exhibit long CoT behaviors, even though understanding and generating such elaborate demonstrations typically require complex abilities. This approach consistently leads to improved task performance. We provided an in-depth analysis that illuminates the extent to which long CoT ICL can enhance performance across different task difficulties and identifies the primary factors driving these improvements. We believe our work represents a crucial initial step towards a deeper understanding of long CoT reasoning and how to effectively elicit these advanced behaviors in LLMs.

# REPRODUCIBILITY STATEMENT

In this study, to ensure the reproducibility of our approach, we provide key information from our submission as follows.

1. **Source Code and Data.** We have submitted the source code of our approach in the supplementary materials.

2. **Experimental Details.** We list the detailed experiment settings, computational resources.

3. **Evaluation and Construct problem-relevant demonstrations.** We provide a detailed evaluation of reasoning ability as well as the algorithms and prompts used to construct problem-relevant demonstrations in the Appendix.

## ETHICS STATEMENT

The authors confirm their adherence to the Code of Ethics. This research is purely methodological and does not involve human subjects or applications with foreseeable negative societal impacts.

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

## THE USE OF LLMs

In the preparation of this manuscript, we employed large language models (LLMs) as a general-purpose writing aid for sentence-level editing, including improving grammar, clarity, and readability. The LLMs did not contribute to any of the core research aspects of this work, such as the formulation of ideas, the design of algorithms, theoretical derivations, or the execution and analysis of experiments. The intellectual content and all claims made within this paper are solely the work of the human authors, who bear full responsibility for the final manuscript.

## A   DETAILED EVALUATION METRICS

### A.1   METRIC

#### A.1.1   LOW-VARIANCE PASS@K METRIC

The metric pass@K reflects the proportion of problems that can be correctly solved within $K$ attempts. Directly estimating pass@K using only $K$ generated answers often incurs high variance, leading to inaccurate results. To mitigate this issue, we follow the unbiased estimation algorithm proposed by Chen et al. (2021). The estimator is defined as:

$$pass@K := \mathbb{E}_{x_i \sim D} \left[ 1 - \frac{\binom{n-c_i}{K}}{\binom{n}{K}} \right] \tag{1}$$

where $c_i$ denotes the number of correct solutions among the $n$ generated samples for problem $x_i \in \mathcal{D}$. The estimator computes the probability that at least one correct solution appears within $K$ attempts by subtracting the probability that all $K$ samples are incorrect. Notably, smaller values of $K$ lead to a more accurate estimation.

#### A.1.2   REASONING BEHAVIOR RATE

To monitor the model's reasoning patterns, we quantify behaviors such as reflection, verification, and correction. Inspired by prior work (Yeo et al., 2025a; Xie et al., 2025), we construct a keyword-based detection system that identifies three categories: **Reflect**, capturing rethinking or exploring alternatives; **Verify**, indicating self-monitoring and re-evaluation; and **Correct**, reflecting error recognition and modification. The average frequency of these behaviors across benchmarks serves as a proxy for assessing reasoning depth, self-monitoring, and correction ability, while also enabling comparisons across models and prompting strategies.

The specific keywords associated with each behavioral category are defined as follows:

- **Reflect**: This category is identified by keywords indicative of re-evaluating the current approach or considering alternatives, such as: `"however"`, `"reflect"`, `"wait"`, `"reconsider"`, `"think again"`, `"rethink"`, and `"alternatively"`.

- **Verify**: This category captures self-monitoring and consistency checks, signaled by keywords including: `"verify"`, `"check"`, `"confirm"`, `"re-evaluate"`, `"reevaluate"`, `"re-examine"`, `"reexamine"`, `"reanalyze"`, and `"recheck"`.

- **Correct**: This category reflects the recognition and amendment of errors, characterized by keywords like: `"correct"`, `"revise"`, and `"adjust"`.

## B   ALGORITHM PIPELINE

### B.1   EVALUATING THE REASONING ABILITY

In this section, we present an evaluation of the quality of long CoT reasoning. The evaluation algorithm is shown in Algorithm 1.

---

**Algorithm 1** Algorithm for Evaluating the Reasoning Ability

---

**Require:** Language model $\pi_\theta$, oracle model $\mathcal{O}$, problem set $S = \{s_1, s_2, \ldots, s_{|S|}\}$, number of trials $N$,

    prompt templates $P_{\text{extract}}$ (reasoning extraction), $P_{\text{eval}}$ (process evaluation)

    **for** each problem $s_i \in S$, $i = 1, 2, \ldots, |S|$ **do**

        **Reference Generation:** Obtain reference answer and extract key reasoning steps:
$$(\hat{y}_i, K_i) = \mathcal{O}(s_i; P_{\text{extract}})$$

        Initialize score collection: $\text{Scores}_i = \{\}$

        **for** trial $j = 1, 2, \ldots, N$ **do**

            **LLM Response:** Generate response from language model: $y_{i,j} = \pi_\theta(s_i)$

            **Reasoning Evaluation:** Score the reasoning process using oracle model and key steps:
$$\text{score}_{i,j} = \mathcal{O}_{\text{eval}}(s_i, y_{i,j}, K_i; P_{\text{eval}})$$

            Add score to collection: $\text{Scores}_i = \text{Scores}_i \cup \{\text{score}_{i,j}\}$

        **end for**

        **Distribution Analysis:** Analyze the score distribution $\text{Scores}_i$ for problem $s_i$

    **end for**

---

**Note:** Prompt templates $P_{\text{extract}}$ and $P_{\text{eval}}$ are detailed in Section C.1.

## B.2 GENERATING PROBLEM-RELEVANT DEMONSTRATIONS

In this section, we present the detailed procedure for generating problem-relevant demonstrations. The algorithm is illustrated in Algorithm 2.

---

**Algorithm 2** Algorithm for Generating Problem-Relevant Demonstrations

---

**Require:** Teacher model $\mathcal{T}$, problem set $S = \{s_1, s_2, \ldots, s_{|S|}\}$,

    prompt templates $P_{\text{extract}}$ (knowledge extraction), $P_{\text{gen}}$ (question generation),

    number of consistency trials $M = 4$, consistency threshold $\tau = 3$

**Ensure:** Problem-demonstration pairs $D = \{(s_i, q_i)\}_{i=1}^{|S|}$

    Initialize demonstration collection: $D = \{\}$

    **for** each problem $s_i \in S$, $i = 1, 2, \ldots, |S|$ **do**

        **Knowledge Extraction:** Solve problem and extract key knowledge points:
$$(y_i, K_i) = \mathcal{T}(s_i; P_{\text{extract}})$$

        Initialize success flag: found = False

        **while** found = False **do**

            **New Question Generation:** Generate related question based on problem and knowledge:
$$q_{\text{candidate}} = \mathcal{T}(s_i, K_i; P_{\text{gen}})$$

            **8-gram Filtering:** Check for uniqueness using 8-gram overlap

            **if** $q_{\text{candidate}}$ passes 8-gram filtering **then**

                Initialize answer collection: Answers = $\{\}$

                **for** trial $j = 1, 2, \ldots, M$ **do**

                    Generate answer: $a_j = \mathcal{T}(q_{\text{candidate}})$

                    Answers = Answers $\cup \{a_j\}$

                **end for**

                **Consistency Check:** Count most frequent answer in Answers

                $\text{max\_count} = \max_a |\{a_j \in \text{Answers} : a_j = a\}|$

                **if** $\text{max\_count} \geq \tau$ **then**

                    $q_i = q_{\text{candidate}}$

                    found = True

                **end if**

            **end if**

        **end while**

        **Save Pair:** $D = D \cup \{(s_i, q_i)\}$

    **end for**

    **return** $D$

---

**Note:** Prompt templates $P_{\text{extract}}$ and $P_{\text{gen}}$ are detailed in Section C.2.

## C  PROMPTS

### C.1  THE PROMPTS FOR QUANTIFY REASONING ABILITY

The prompt for extracting the key intermediate steps and evaluating the error steps are shown in the following. Specifically, the prompt of extracting the key intermediate steps is adapted from Jiang et al. (2025).

---

**Reasoning Step Extraction Prompt Template**

Analyze the provided reasoning text and extract a strictly ordered, atomic sequence of key reasoning steps. Focus on extracting the validated, logically essential progression of thoughts while excluding backtracking, rechecks, or redundant details.

**EXTRACTION RULES:**

1. **Logical Flow Identification**: Find the key steps and the logical flow of reasoning

2. **Atomic Requirement**: Each step must represent a single, indivisible logical action that directly advances the reasoning

3. **Redundancy Elimination**: Determine the correct version of the step, ignoring redundant information. A correct step should be able to push the reasoning logic forward and have no errors in itself

4. **Completeness Guarantee**: Do not skip steps. Do not merge steps. Use the original phrasing where possible

5. **Verification Filter**: Do not include verification steps unless it introduces new constraints

6. **Sequential Organization**: Organize the steps into a coherent sequence of key reasoning steps and number it sequentially (1., 2., 3., ...)

7. **Format Compliance**: Maintain strict output format

**EXCLUSIONS:**

- Backtracking processes
- Rechecking steps
- Redundant details

**STANDARD OUTPUT FORMAT:**
<reasoning_process>
Step 1. [concise statement]: [Detail step]
Step 2. [concise statement]: [Detail step]
Step 3. [concise statement]: [Detail step]
· · ·
</reasoning_process>
**USAGE INSTRUCTIONS:**
This template uses the format method to fill in the specific reasoning_text parameter, generating complete extraction instructions. The extraction results can be used for downstream tasks such as reasoning process analysis, step evaluation, and logical chain verification.
**APPLICATION SCENARIOS:**
Mathematical reasoning analysis, logical deduction verification, problem-solving process evaluation, reasoning quality assessment, and other scenarios requiring key step identification and extraction.

---

**Process Score Evaluation Prompt Template**

You are an expert mathematics teacher evaluating a student's solution to a math competition problem. Your task is to assign a process score based on the student's reasoning process, even if their final answer is incorrect.

**PROBLEM:**
{question}
**STANDARD KEY STEPS:**
{key_steps}
**STUDENT'S ANSWER:**
{answer}
**SCORING CRITERIA:**
The process score ranges from 0 to 1 and consists of three components:

1. **Problem Understanding (0.1 points)**: Does the student correctly understand what the problem is asking for?

- Award 0.1 if they understand the problem correctly

- Award 0 if they misunderstand the problem

2. **Approach Direction (0.1 points)**: Is the student's overall approach/method appropriate for solving this problem?

- Award 0.1 if their approach is generally correct or reasonable

- Award 0 if their approach is fundamentally wrong

3. **Step Execution (0.8 points)**: How many of the key reasoning steps did the student execute correctly?

- Calculate: (Number of correctly executed steps / Total number of key steps) × 0.8

- Count partial credit for steps that are attempted but have minor errors

**EVALUATION INSTRUCTIONS:**

1. Compare the student's reasoning against the standard key steps

2. Identify which key steps the student successfully completed (even if not in the exact same order)

3. Give partial credit for steps that show correct reasoning but may have minor computational errors

4. Focus on the reasoning process, not just the final answer

**OUTPUT FORMAT:**
<result>
Problem Understanding: [0 or 0.1] - [Brief explanation]
Approach Direction: [0 or 0.1] - [Brief explanation]
Step Execution: [X/Y steps correct] = [score out of 0.8] - [List which steps were done correctly]
Total Process Score: [sum of above three components]
</result>
Please evaluate the student's solution carefully and provide your scoring.

## C.2 THE PROMPTS FOR GENERATING RELATED QUESTIONS

The prompt for generating related problems for each problem is shown in the following. Additionally, we provide examples of the final input to the model under two conditions: one with randomly selected problems and the other with constructed related problems.

## Knowledge Extraction Prompt Template

You are an expert in mathematics and knowledge extraction.
Your task is to process a math problem and its answer in three stages:
**PROCESSING STAGES:**

1. **Solution Derivation**: Carefully solve the given problem step by step, showing the reasoning that leads to the final answer

2. **Knowledge Extraction**: Based on the reasoning process, identify:
   - The key **concepts** involved (mathematical ideas, topics, or theories)
   - The **skills** required (specific techniques or problem-solving methods)
   - The **theorems** or mathematical results explicitly or implicitly used in the solution

3. **Output Format**: Present the extracted knowledge in the following strict JSON format only

---

**Problem**:
{original_question}
**Given Answer**:
{original_answer}

---

**EXECUTION SEQUENCE:**
Now, begin with the solution derivation, then perform the knowledge extraction, and finally output only the JSON object for extracted_knowledge.
**STANDARD OUTPUT FORMAT:**

```
{
  "concepts": [...],
  "skills": [...],
  "theorems": [...]
}
```

---

### New Question Generation Prompt Template

You are a mathematics education expert.
Your task is to design a **new practice question** that will help a student who could not solve the following original problem eventually solve it.

---

**Original Problem**:
{original_question}
**Original Answer**:
{original_answer}
**Extracted Knowledge (concepts, skills, theorems)**:
{extracted_knowledge}

---

**REQUIREMENTS:**

1. **Problem Analysis**: Carefully analyze the original problem and the extracted knowledge

2. **New Question Design**: Create a **new_question** that:
   - Trains the same concepts, skills, and theorems as required in the original problem
   - Is easier or more guided than the original problem, serving as a stepping stone
   - Is self-contained and solvable without referring to the original problem

3. **Explanation Requirement**: After creating the new_question, briefly explain in one or two sentences how solving it prepares the student to solve the original problem

**CONSTRAINTS TO AVOID CHEATING:**

- The new_question must **not be too similar in surface wording or structure** to the original problem
- **Specifically avoid**:
  - Copying long phrases or expressions directly
  - Keeping the same problem type with only small number changes
- **Instead**: Change the **problem framing**, **question type**, or **context**, while ensuring that the underlying knowledge being practiced remains aligned with the original problem

**OUTPUT FORMAT:**
Respond in strict JSON format:

```
{
   "new_question": "<the designed practice problem statement>",
   "rationale": "<how this helps prepare for solving the original
      problem>"
}
```

The following examples demonstrate the final model inputs under two different demonstration selection strategies for AIME25 mathematical problems. The first shows related problem selection where demonstration examples share similar mathematical concepts, while the second shows random problem selection with diverse demonstration topics.

972
973
974
975
976
977
978
979
980
981
982
983
984
985
986
987
988
989
990
991
992
993
994
995
996
997
998
999
1000
1001
1002
1003
1004
1005
1006
1007
1008
1009
1010
1011
1012
1013
1014
1015
1016
1017
1018
1019
1020
1021
1022
1023
1024
1025

## Final Inputs Example

**Related Problem Selection Input:**
You are a mathematical problem solver.
Below are 2 examples of how to solve mathematical problems. Study these examples carefully to understand the problem-solving approach and reasoning patterns.
**DEMONSTRATION EXAMPLE 1:**
**Problem:** Let O = (0,0). Let X = (30,0) and Y = (0,H) for some positive H. On segment OX take points U and V so that OU:UV:VX = 1:3:2. On segment OY take points W and Z so that OW:WZ:ZY = 2:6:2. Let U' be the reflection of U across W (so W is the midpoint of UU') and let Z' be the reflection of Z across V (so V is the midpoint of ZZ'). The quadrilateral with vertices U, V, Z, W taken in that cyclic order has area 240. Find the area of the hexagon with vertices O, U', X, V, Y, Z' taken in that order. Give your final answer in a box. Please reason step by step, and put your final answer within \boxed{}.
**Solution:** {Solution 1}
**DEMONSTRATION EXAMPLE 2:**
**Problem:** Let a and b be two noncollinear vectors in the plane and let parallelogram OABC be formed by O (the origin), A = a, B = b, and C = a + b. Points P and Q lie on OA and OB respectively with OP = (1/4)·OA and OQ = (1/3)·OB. Let P' be the reflection of P across the midpoint of segment OA, and let Q' be the reflection of Q across the midpoint of segment OB. If the area of quadrilateral PP'Q'Q is 60, find the area of parallelogram OABC (i.e. find —a × b—). (You may use that reflecting X about a point R gives X' = 2R - X and that area is bilinear in the side vectors.) Please reason step by step, and put your final answer within \boxed{}.
**Solution:** {Solution 2}
**YOUR TASK:**
Now, solve the following problem by applying the reasoning skills and solution patterns demonstrated in the examples above:
**Problem:** On $\triangle ABC$ points $A, D, E,$ and $B$ lie that order on side $\overline{AB}$ with $AD = 4, DE = 16,$ and $EB = 8$. Points $A, F, G,$ and $C$ lie in that order on side $\overline{AC}$ with $AF = 13, FG = 52,$ and $GC = 26$. Let $M$ be the reflection of $D$ through $F$, and let $N$ be the reflection of $G$ through $E$. Quadrilateral $DEGF$ has area 288. Find the area of heptagon $AFNBCEM$. Let's think step by step and output the final answer within \boxed{}. Please reason step by step, and put your final answer within \boxed{}.
**Solution:**

**Random Problem Selection Input:**
You are a mathematical problem solver.
Below are 2 examples of how to solve mathematical problems. Study these examples carefully to understand the problem-solving approach and reasoning patterns.
**DEMONSTRATION EXAMPLE 1:**
**Problem:** On the AMC 8 contest Billy answers 13 questions correctly, answers 7 questions incorrectly and doesn't answer the last 5. What is his score? Please reason step by step, and put your final answer within \boxed{}.
**Solution:** {Solution 1}
**DEMONSTRATION EXAMPLE 2:**
**Problem:** If $a(x + 1) = x^3 + 3x^2 + 3x + 1$, find $a$ in terms of $x$.
If $a - 1 = 0$, then the value of $x$ is 0 or $b$. What is $b$ ?
If $pc^4 = 32, pc = b^2$, and $c$ is positive, what is the value of $c$ ?
$P$ is an operation such that $P(A \cdot B) = P(A) + P(B)$.
$P(A) = y$ means $A = 10^y$. If $d = A \cdot B, P(A) = 1$ and $P(B) = c$, find $d$. Please reason step by step, and put your final answer within \boxed{}.
**Solution:** {Solution2}
**YOUR TASK:**
Now, solve the following problem by applying the reasoning skills and solution patterns demonstrated in the examples above:
**Problem:** On $\triangle ABC$ points $A, D, E,$ and $B$ lie that order on side $\overline{AB}$ with $AD = 4, DE = 16,$ and $EB = 8$. Points $A, F, G,$ and $C$ lie in that order on side $\overline{AC}$ with $AF = 13, FG = 52,$ and $GC = 26$. Let $M$ be the reflection of $D$ through $F$, and let $N$ be the reflection of $G$ through $E$. Quadrilateral $DEGF$ has area 288. Find the area of heptagon $AFNBCEM$. Let's think step by step and output the final answer within \boxed{}. Please reason step by step, and put your final answer within \boxed{}.
**Solution:**

# D    MORE EXPERIMENTS

## D.1    SOLUTIONS OF ICL AND BASE MODEL

This section presents the answers on AIME25 obtained by Qwen2.5-32B through direct answering and through ICL with long CoT demonstrations. An asterisk preceding a method name indicates that the answer is correct. From these answers, we can observe the differences between using ICL and direct answering.

---

**Problem 4**

**Problem:** Six points $A, B, C, D, E$ and $F$ lie in a straight line in that order. Suppose that $G$ is a point not on the line and that $AC = 26$, $BD = 22$, $CE = 31$, $DF = 33$, $AF = 73$, $CG = 40$, and $DG = 30$. Find the area of $\triangle BGE$.

**Base Model*:** To solve this problem, we need to use the given information to find the co-ordinates of the points and then use the formula for the area of a triangle given its vertices. Let's assume that point $A$ is at the origin, so $A = (0, 0)$. Since all points lie on a straight line in the order $A, B, C, D, E, F$, we can represent the coordinates of the points as follows: $A = (0, 0)$ $B = (x_1, 0)$ $C = (x_2, 0)$ $D = (x_3, 0)$ $E = (x_4, 0)$ $F = (x_5, 0)$ Now, let's use the given distances to find the coordinates of the points: 1. $AC = 26 \Rightarrow x_2 = 26$ 2. $BD = 22 \Rightarrow x_3 - x_1 = 22$ 3. $CE = 31 \Rightarrow x_4 - x_2 = 31 \Rightarrow x_4 = 57$ 4. $DF = 33 \Rightarrow x_5 - x_3 = 33$ 5. $AF = 73 \Rightarrow x_5 = 73$ Now we can find the coordi-nates of the remaining points: 1. From $x_5 = 73$ and $x_5 - x_3 = 33$, we get $x_3 = 40$. 2. From $x_3 - x_1 = 22$, we get $x_1 = 18$. So, the coordinates of the points are: $A = (0, 0)$ $B = (18, 0)$ $C = (26, 0)$ $D = (40, 0)$ $E = (57, 0)$ $F = (73, 0)$ Now, we need to find the coordinates of point $G$. We are given that $CG = 40$ and $DG = 30$. Let's assume that $G = (x, y)$. Then, we have the following equations: 1. $CG = 40 \Rightarrow \sqrt{(x - 26)^2 + y^2} = 40$ 2. $DG = 30 \Rightarrow \sqrt{(x - 40)^2 + y^2} = 30$ Squaring both equations and subtracting, we get: $(x - 26)^2 - (x - 40)^2 = 40^2 - 30^2$ Solving this equation, we get $x = 31$. Substitut-ing this value back into either equation, we get $y = 24$. So, the coordinates of point $G$ are $(31, 24)$. Now, we can use the formula for the area of a triangle given its vertices: $Area = \frac{1}{2}|x_1(y_2 - y_3) + x_2(y_3 - y_1) + x_3(y_1 - y_2)|$ Substituting the coordinates of points $B$, $G$, and $E$ into the formula, we get: $Area = \frac{1}{2}|18(24 - 0) + 31(0 - 0) + 57(0 - 24)| = \frac{1}{2}|18(24) - 57(24)| = \frac{1}{2}|24(18 - 57)| = \frac{1}{2}|24(-39)| = \frac{1}{2} \cdot 24 \cdot 39 = \boxed{468}$

**ICL:** $< think >$ Alright, let's break this down step by step. We have six points $A, B, C, D, E$ and $F$ lying in a straight line in that order. We also have a point $G$ not on the line, and we know the lengths of several line segments. We need to find the area of $\triangle BGE$. First, let's find the lengths of $AB, BC, CD, DE$, and $EF$. We can use the given information to do this: 1. $AC = AB + BC = 26$ 2. $BD = BC + CD = 22$ 3. $CE = CD + DE = 31$ 4. $DF = DE + EF = 33$ 5. $AF = AB + BC + CD + DE + EF = 73$ We can solve these equations to find the lengths of $AB, BC, CD, DE$, and $EF$: 1. From $AC = 26$, we have $AB + BC = 26$. 2. From $BD = 22$, we have $BC + CD = 22$. 3. From $CE = 31$, we have $CD + DE = 31$. 4. From $DF = 33$, we have $DE + EF = 33$. 5. From $AF = 73$, we have $AB + BC + CD + DE + EF = 73$. Substituting the values of $AC, BD, CE$, and $DF$ into the equations, we get: 1. $AB + BC = 26$ 2. $BC + CD = 22$ 3. $CD + DE = 31$ 4. $DE + EF = 33$ 5. $AB + BC + CD + DE + EF = 73$ Solving these equations, we find: 1. $AB = 10$ 2. $BC = 16$ 3. $CD = 6$ 4. $DE = 25$ 5. $EF = 8$ Now, let's find the coordinates of points $A, B, C, D, E, F$ and $G$. We can assume that $A$ is at the origin, so $A = (0, 0)$. Then, we have: 1. $B = (10, 0)$ 2. $C = (26, 0)$ 3. $D = (32, 0)$ 4. $E = (57, 0)$ 5. $F = (65, 0)$ Since $G$ is not on the line, we can assume that $G$ has coordinates $(x, y)$. We know that $CG = 40$ and $DG = 30$, so we can use the distance formula to find the coordinates of $G$: 1. $CG = \sqrt{(x - 26)^2 + y^2} = 40$ 2. $DG = \sqrt{(x - 32)^2 + y^2} = 30$ Solving these equations, we find that $G = (26, 30)$.

---

Now, we can find the area of $\triangle BGE$ using the coordinates of $B, G, E$: 1. $B = (10, 0)$ 2. $G = (26, 30)$ 3. $E = (57, 0)$ Using the formula for the area of a triangle with coordinates $(x_1, y_1), (x_2, y_2), (x_3, y_3)$: Area $= \frac{1}{2}|x_1(y_2 - y_3) + x_2(y_3 - y_1) + x_3(y_1 - y_2)|$ Substituting the coordinates of $B, G, E$ into the formula, we get: Area $= \frac{1}{2}|10(30 - 0) + 26(0 - 0) + 57(0 - 30)|$ Simplifying, we find: Area $= \frac{1}{2}|300 + 0 - 1710|$ Area $= \frac{1}{2}|-1410|$ Area $= \frac{1}{2} \cdot 1410$ Area $= \boxed{705}$ $</think>$ First, we find the lengths of $AB$, $BC$, $CD$, $DE$, and $EF$ using the given information. Then, we find the coordinates of points $A, B, C, D, E, F$, and $G$. Finally, we use the coordinates of $B, G, E$ to find the area of $\triangle BGE$ using the formula for the area of a triangle with coordinates. The area of $\triangle BGE$ is $\boxed{705}$.

---

## Problem 7

**Problem:** Let $k$ be real numbers such that the system $|25 + 20i - z| = 5$ and $|z - 4 - k| = |z - 3i - k|$ has exactly one complex solution $z$. The sum of all possible values of $k$ can be written as $\frac{m}{n}$, where $m$ and $n$ are relatively prime positive integers. Find $m + n$. Here $i = \sqrt{-1}$.

**Base Model:** The first equation represents a circle centered at $25 + 20i$ with radius 5. The second equation represents the perpendicular bisector of the segment joining $4 + k$ and $3i + k$. For there to be exactly one solution, the line must be tangent to the circle. The distance between the center of the circle and the line is the radius of the circle. The equation of the line is $y - x = k$. The distance between the center of the circle and the line is $\frac{|20 - 25 - k|}{\sqrt{2}} = 5$. Solving for $k$ gives $k = -15$ or $k = -5$. The sum of all possible values of $k$ is $-20$, which can be written as $\frac{-40}{2}$. The final answer is $\boxed{42}$.

**ICL\*:** The problem asks us to find the sum of all possible values of $k$ such that the system of equations $|25 + 20i - z| = 5$ and $|z - 4 - k| = |z - 3i - k|$ has exactly one complex solution $z$. To solve this problem, we need to analyze the two equations separately. The first equation represents a circle in the complex plane with center $25 + 20i$ and radius 5. The second equation represents the set of points $z$ that are equidistant from $4 + k$ and $3i + k$, which forms a perpendicular bisector of the line segment joining $4 + k$ and $3i + k$. For the system to have exactly one complex solution, the circle and the line must intersect at exactly one point, which means the line must be tangent to the circle. The center of the circle is $25 + 20i$ and its radius is 5. The line is the perpendicular bisector of the line segment joining $4 + k$ and $3i + k$, which has a midpoint of $\frac{(4+k)+(3i+k)}{2} = 2 + \frac{3i}{2} + k$. The slope of the line segment joining $4 + k$ and $3i + k$ is $\frac{3i - 0}{0 - 4} = -\frac{3}{4}$, so the slope of the perpendicular bisector is $\frac{4}{3}$. The equation of the line can be written as $y - \frac{3}{2} = \frac{4}{3}(x - 2 - k)$, which simplifies to $4x - 3y = 8 + 4k - \frac{9}{2}$. The distance from the center of the circle to the line must be equal to the radius of the circle, which is 5. Using the formula for the distance from a point to a line, we have:

$$\frac{|4(25) - 3(20) - (8 + 4k - \frac{9}{2})|}{\sqrt{4^2 + (-3)^2}} = 5$$

. Simplifying this equation, we get:

$$|32 - 4k + \frac{9}{2}| = 25$$

. Solving this equation for $k$, we get two possible values of $k$: 1. $32 - 4k + \frac{9}{2} = 25$ 2. $32 - 4k + \frac{9}{2} = -25$. Solving these equations, we find: 1. $k = \frac{23}{8}$ 2. $k = \frac{123}{8}$. The sum of all possible values of $k$ is $\frac{23}{8} + \frac{123}{8} = \frac{146}{8} = \frac{73}{4}$. Therefore, the sum of all possible values of $k$ can be written as $\frac{73}{4}$, where 73 and 4 are relatively prime positive integers. The final answer is $73 + 4 = \boxed{77}$.

## Problem 8

**Problem:**
The parabola with equation $y = x^2 - 4$ is rotated $60°$ counterclockwise around the origin. The unique point in the fourth quadrant where the original parabola and its image intersect has $y$-coordinate $\frac{a - \sqrt{b}}{c}$, where $a$, $b$, and $c$ are positive integers, and $a$ and $c$ are relatively prime. Find $a + b + c$.

**Base Model:** To find the unique point in the fourth quadrant where the original parabola and its image intersect after a 60-degree counterclockwise rotation around the origin, we can follow these steps: 1. **Rotate the parabola:** The equation of the parabola after a 60-degree counterclockwise rotation can be found using the rotation transformation formulas: $x' = x\cos(60°) - y\sin(60°)$ and $y' = x\sin(60°) + y\cos(60°)$. Substituting $\cos(60°) = \frac{1}{2}$ and $\sin(60°) = \frac{\sqrt{3}}{2}$, we get $x' = \frac{x}{2} - \frac{y\sqrt{3}}{2}$ and $y' = \frac{x\sqrt{3}}{2} + \frac{y}{2}$. 2. **Find the intersection:** The intersection of the original parabola and its rotated image can be found by solving the system of equations $y = x^2 - 4$ and $y' = (x')^2 - 4$. 3. **Substitute the rotated coordinates:** Substituting the rotated coordinates into the second equation, we get $\frac{x\sqrt{3}}{2} + \frac{y}{2} = \left(\frac{x}{2} - \frac{y\sqrt{3}}{2}\right)^2 - 4$. 4. **Solve the system of equations:** Solving the system of equations, we find the unique point in the fourth quadrant where the original parabola and its image intersect. 5. **Find the y-coordinate:** The y-coordinate of the unique point in the fourth quadrant is $\frac{a - \sqrt{b}}{c}$, where $a$, $b$, and $c$ are positive integers, and $a$ and $c$ are relatively prime. We can find $a$, $b$, and $c$ by analyzing the solution. Let's implement this in Python using sympy:

Listing 1: Code Hallucination

```python
import sympy as sp

# Define variables
x, y = sp.symbols('x y')

# Define the original parabola equation
original_parabola = y - (x**2 - 4)

# Define the rotated parabola equation
x_prime = x/2 - y*sp.sqrt(3)/2
y_prime = x*sp.sqrt(3)/2 + y/2
rotated_parabola = y_prime - (x_prime**2 - 4)

# Solve the system of equations
solutions = sp.solve([original_parabola, rotated_parabola], (x, y))

# Find the unique point in the fourth quadrant
for sol in solutions:
    if sol[0] > 0 and sol[1] < 0:
        y_coordinate = sol[1]
        break

# Simplify the y-coordinate to the form (a - sqrt(b))/c
y_coordinate_simplified = sp.simplify(y_coordinate)
a, b, c = sp.Wild('a'), sp.Wild('b'), sp.Wild('c')
match = y_coordinate_simplified.match((a - sp.sqrt(b)) / c)
a_val = match[a]
b_val = match[b]
c_val = match[c]

# Calculate a + b + c
result = a_val + b_val + c_val
print(result)
```

The output is 17. The unique point in the fourth quadrant where the original parabola and its image intersect has $y$-coordinate $\frac{a-\sqrt{b}}{c}$, where $a$, $b$, and $c$ are positive integers, and $a$ and $c$ are relatively prime. We found that $a + b + c = \boxed{17}$.

**ICL\*:** $<think>$ Alright, let's tackle this step by step. We're dealing with a parabola and its image after a 60-degree counterclockwise rotation around the origin. We're looking for the y-coordinate of the unique intersection point in the fourth quadrant. First, let's write down the equation of the original parabola: $y = x^2 - 4$. Now, let's find the equation of the rotated parabola. To do this, we'll use the rotation transformation formulas: $x' = x\cos(\theta) - y\sin(\theta)$ $y' = x\sin(\theta) + y\cos(\theta)$ In our case, $\theta = 60°$, so $\cos(60°) = \frac{1}{2}$ and $\sin(60°) = \frac{\sqrt{3}}{2}$. Substituting these values into the transformation formulas, we get: $x' = \frac{x}{2} - \frac{\sqrt{3}y}{2}$ $y' = \frac{\sqrt{3}x}{2} + \frac{y}{2}$ Now, let's substitute $y = x^2 - 4$ into the second equation: $y' = \frac{\sqrt{3}x}{2} + \frac{x^2-4}{2}$ Simplifying this expression, we get: $y' = \frac{x^2+\sqrt{3}x-4}{2}$ Now we have the equation of the rotated parabola: $y' = \frac{x^2+\sqrt{3}x-4}{2}$. Next, we need to find the intersection point of the original parabola and its image in the fourth quadrant. To do this, we'll set the equations of the two parabolas equal to each other: $x^2 - 4 = \frac{x^2+\sqrt{3}x-4}{2}$ Multiplying both sides by 2 to eliminate the fraction, we get: $2x^2 - 8 = x^2 + \sqrt{3}x - 4$ Rearranging the terms, we have: $x^2 - \sqrt{3}x - 4 = 0$ Now, we'll solve this quadratic equation using the quadratic formula: $x = \frac{-b\pm\sqrt{b^2-4ac}}{2a}$ In our case, $a = 1$, $b = -\sqrt{3}$, and $c = -4$. Substituting these values into the formula, we get: $x = \frac{\sqrt{3}\pm\sqrt{3+16}}{2}$ Simplifying this expression, we have: $x = \frac{\sqrt{3}\pm\sqrt{19}}{2}$ Since we're looking for the intersection point in the fourth quadrant, we'll choose the negative root: $x = \frac{\sqrt{3}-\sqrt{19}}{2}$ Now, let's find the corresponding y-coordinate by substituting this x-value into the equation of the original parabola: $y = x^2 - 4$ $y = \left(\frac{\sqrt{3}-\sqrt{19}}{2}\right)^2 - 4$ Simplifying this expression, we get: $y = \frac{3-2\sqrt{57}+19-16}{4}$ $y = \frac{6-2\sqrt{57}}{4}$ Finally, we can simplify the expression further: $y = \frac{3-\sqrt{57}}{2}$ Now we have the y-coordinate of the intersection point in the fourth quadrant: $\frac{3-\sqrt{57}}{2}$. We're given that $a$, $b$, and $c$ are positive integers, and $a$ and $c$ are relatively prime. So, $a = 3$, $b = 57$, and $c = 2$. Therefore, $a + b + c = 3 + 57 + 2 = 62$. So, the final answer is: $\boxed{62}$ $</think>$ To find the y-coordinate of the intersection point in the fourth quadrant, we first find the equation of the rotated parabola using the rotation transformation formulas. Then, we set the equations of the original and rotated parabolas equal to each other and solve the resulting quadratic equation. Finally, we substitute the x-coordinate of the intersection point into the equation of the original parabola to find the y-coordinate. The y-coordinate of the intersection point is $\frac{3-\sqrt{57}}{2}$. Therefore, $a = 3$, $b = 57$, and $c = 2$, and the final answer is $a + b + c = 3 + 57 + 2 = \boxed{62}$

---

### Problem 15

**Problem:** Six points $A, B, C, D, E$ and $F$ lie in a straight line in that order. Suppose that $G$ is a point not on the line and that $AC = 26$, $BD = 22$, $CE = 31$, $DF = 33$, $AF = 73$, $CG = 40$, and $DG = 30$. Find the area of $\triangle BGE$.

**Base Model*:** To solve this problem, we need to use the given information to find the coordinates of the points and then use the formula for the area of a triangle given its vertices. Let's assume that point $A$ is at the origin, so $A = (0,0)$. Since all points lie on a straight line in the order $A, B, C, D, E, F$, we can represent the coordinates of the points as follows: $A = (0,0)$, $B = (x_1, 0)$, $C = (x_2, 0)$, $D = (x_3, 0)$, $E = (x_4, 0)$, $F = (x_5, 0)$. Now, let's use the given distances to find the coordinates of the points: 1. $AC = 26 \Rightarrow x_2 = 26$ 2. $BD = 22 \Rightarrow x_3 - x_1 = 22$ 3. $CE = 31 \Rightarrow x_4 - x_2 = 31 \Rightarrow x_4 = 574$. $DF = 33 \Rightarrow x_5 - x_3 = 335$. $AF = 73 \Rightarrow x_5 = 73$. Now we can find the coordinates of the remaining points: 1. From $x_5 = 73$ and $x_5 - x_3 = 33$, we get $x_3 = 40$. 2. From $x_3 - x_1 = 22$, we get $x_1 = 18$. So, the coordinates of the points are: $A = (0,0)$, $B = (18, 0)$, $C = (26, 0)$, $D = (40, 0)$, $E = (57, 0)$, $F = (73, 0)$. Now, we need to find the coordinates of point $G$. We are given that $CG = 40$ and $DG = 30$. Let's assume that $G = (x, y)$. Then, we have the following equations: 1. $CG = 40 \Rightarrow \sqrt{(x - 26)^2 + y^2} = 40$ 2. $DG = 30 \Rightarrow \sqrt{(x - 40)^2 + y^2} = 30$. Squaring both equations and subtracting, we get: $(x - 26)^2 - (x - 40)^2 = 40^2 - 30^2$. Solving this equation, we get $x = 31$. Substituting this value back into either equation, we get $y = 24$. So, the coordinates of point $G$ are $(31, 24)$. Now, we can use the formula for the area of a triangle given its vertices: $Area = \frac{1}{2}|x_1(y_2 - y_3) + x_2(y_3 - y_1) + x_3(y_1 - y_2)|$. Substituting the coordinates of points $B$, $G$, and $E$ into the formula, we get: $Area = \frac{1}{2}|18(24 - 0) + 31(0 - 0) + 57(0 - 24)| = \frac{1}{2}|18(24) - 57(24)| = \frac{1}{2}|24(18 - 57)| = \frac{1}{2}|24(-39)| = \frac{1}{2} \cdot 24 \cdot 39 = \boxed{468}$

**ICL:** Alright, let's break this down step by step. We have six points $A, B, C, D, E$ and $F$ lying in a straight line in that order. We also have a point $G$ not on the line, and we know the lengths of several line segments. We need to find the area of $triangle BGE$. First, let's find the lengths of $AB, BC, CD, DE$, and $EF$. We can use the given information to do this: 1. $AC = AB + BC = 26$ 2. $BD = BC + CD = 22$ 3. $CE = CD + DE = 31$ 4. $DF = DE + EF = 33$ 5. $AF = AB + BC + CD + DE + EF = 73$ We can solve these equations to find the lengths of $AB, BC, CD, DE$, and $EF$: 1. From $AC = 26$, we have $AB + BC = 26$. 2. From $BD = 22$, we have $BC + CD = 22$. 3. From $CE = 31$, we have $CD + DE = 31$. 4. From $DF = 33$, we have $DE + EF = 33$. 5. From $AF = 73$, we have $AB + BC + CD + DE + EF = 73$. Substituting the values of $AC, BD, CE$, and $DF$ into the equations, we get: 1. $AB + BC = 26$ 2. $BC + CD = 22$ 3. $CD + DE = 31$ 4. $DE + EF = 33$ 5. $AB + BC + CD + DE + EF = 73$. Solving these equations, we find: 1. $AB = 10$ 2. $BC = 16$ 3. $CD = 6$ 4. $DE = 25$ 5. $EF = 8$. Now, let's find the coordinates of points $A, B, C, D, E, F$ and $G$. We can assume that $A$ is at the origin, so $A = (0,0)$. Then, we have: 1. $B = (10, 0)$ 2. $C = (26, 0)$ 3. $D = (32, 0)$ 4. $E = (57, 0)$ 5. $F = (65, 0)$. Since $G$ is not on the line, we can assume that $G$ has coordinates $(x, y)$. We know that $CG = 40$ and $DG = 30$, so we can use the distance formula to find the coordinates of $G$: 1. $CG = \sqrt{(x - 26)^2 + y^2} = 40$ 2. $DG = \sqrt{(x - 32)^2 + y^2} = 30$. Solving these equations, we find that $G = (26, 30)$. Now, we can find the area of $\triangle BGE$ using the coordinates of $B, G, E$: 1. $B = (10, 0)$ 2. $G = (26, 30)$ 3. $E = (57, 0)$. Using the formula for the area of a triangle with coordinates $(x_1, y_1), (x_2, y_2), (x_3, y_3)$: Area $= \frac{1}{2}|x_1(y_2 - y_3) + x_2(y_3 - y_1) + x_3(y_1 - y_2)|$. Substituting the coordinates of $B, G, E$ into the formula, we get: $textArea = \frac{1}{2}|10(30 - 0) + 26(0 - 0) + 57(0 - 30)|$ Simplifying, we find: Area $= \frac{1}{2}|300 + 0 - 1710|$. Area $= \frac{1}{2}| - 1410|$. Area $= \frac{1}{2} \cdot 1410$. Area $= \boxed{705}$

Problem 27

**Problem:** Let $x_1, x_2, x_3, \ldots$ be a sequence of rational numbers defined by $x_1 = \frac{25}{11}$ and

$$x_{k+1} = \frac{1}{3} \left( x_k + \frac{1}{x_k} - 1 \right)$$

for all $k \geq 1$. Then $x_{2025}$ can be expressed as $\frac{m}{n}$, where $m$ and $n$ are relatively prime positive integers. Find the remainder when $m + n$ is divided by 1000.

**Base Model:** To solve this problem, we need to analyze the given recursive sequence and find a pattern or a closed-form expression for $x_k$. Let's start by computing the first few terms of the sequence to see if we can identify any patterns. 1. Compute $x_1$:

$$x_1 = \frac{25}{11}$$

2. Compute $x_2$:

$$x_2 = \frac{1}{3} \left( x_1 + \frac{1}{x_1} - 1 \right) = \frac{1}{3} \left( \frac{25}{11} + \frac{11}{25} - 1 \right) = \frac{157}{275}$$

3. Compute $x_3$:

$$x_3 = \frac{1}{3} \left( x_2 + \frac{1}{x_2} - 1 \right) = \frac{1}{3} \left( \frac{157}{275} + \frac{275}{157} - 1 \right) = \frac{18983}{43325}$$

4. Compute $x_4$:

$$x_4 = \frac{1}{3} \left( x_3 + \frac{1}{x_3} - 1 \right) = \frac{1}{3} \left( \frac{18983}{43325} + \frac{43325}{18983} - 1 \right) = \frac{155683843}{80840025}$$

Now, let's analyze the pattern in the sequence. We can see that the numerators and denominators of the terms are increasing rapidly. However, we can notice that the numerators and denominators of consecutive terms are relatively prime. This suggests that the terms of the sequence are in their simplest form. To find $x_{2025}$, we need to find a pattern in the sequence or a closed-form expression for $x_k$. However, finding a closed-form expression for $x_k$ is not straightforward. Instead, we can use the fact that the terms of the sequence are in their simplest form and that the numerators and denominators are relatively prime. Let's denote the numerator of $x_k$ as $a_k$ and the denominator as $b_k$. Then, we have:

$$x_k = \frac{a_k}{b_k}$$

We can rewrite the recursive formula as:

$$\frac{a_{k+1}}{b_{k+1}} = \frac{1}{3} \left( \frac{a_k}{b_k} + \frac{b_k}{a_k} - 1 \right)$$

Multiplying both sides by $3a_k b_k$, we get:

$$3a_{k+1} b_k = a_k^2 + b_k^2 - a_k b_k$$

This equation shows that $a_{k+1}$ and $b_{k+1}$ are relatively prime, as $a_k$ and $b_k$ are relatively prime. Now, let's find the remainder when $m + n$ is divided by 1000. We know that $x_{2025}$ can be expressed as $\frac{m}{n}$, where $m$ and $n$ are relatively prime positive integers. Therefore, we need to find the remainder when $m + n$ is divided by 1000. Since $a_{k+1}$ and $b_{k+1}$ are relatively prime, we can conclude that $m$ and $n$ are relatively prime. Therefore, we can use the Chinese Remainder Theorem to find the remainder when $m + n$ is divided by 1000. However, finding the exact values of $m$ and $n$ is not possible without a closed-form expression for $x_k$. Therefore, we can only conclude that the remainder when $m + n$ is divided by 1000 is a number between 0 and 999. The answer is $\boxed{0}$

**ICL:** $<think>$ Okay, so I need to find $x_{2025}$ for a sequence of rational numbers defined by $x_1 = \frac{25}{11}$ and $x_{k+1} = \frac{1}{3} \left( x_k + \frac{1}{x_k} - 1 \right)$ for all $k \geq 1$. Then, I need to express $x_{2025}$ as $\frac{m}{n}$,

where $m$ and $n$ are relatively prime positive integers. Finally, I need to find the remainder when $m + n$ is divided by 1000. First, let's find a pattern in the sequence. I'll compute the first few terms: $x_1 = \frac{25}{11}$ $x_2 = \frac{1}{3}\left(\frac{25}{11} + \frac{11}{25} - 1\right) = \frac{1}{3}\left(\frac{625+121-275}{275}\right) = \frac{1}{3}\left(\frac{471}{275}\right) = \frac{157}{275}$ $x_3 = \frac{1}{3}\left(\frac{157}{275} + \frac{275}{157} - 1\right) = \frac{1}{3}\left(\frac{24649+75625-43225}{43225}\right) = \frac{1}{3}\left(\frac{57049}{43225}\right) = \frac{57049}{129675}$ $x_4 = \frac{1}{3}\left(\frac{57049}{129675} + \frac{129675}{57049} - 1\right) = \frac{1}{3}\left(\frac{3254000001+1682705625-756250000}{756250000}\right) = \frac{1}{3}\left(\frac{4180455626}{756250000}\right) = \frac{4180455626}{2268750000}$ Now, I can see that the numerator and denominator of $x_k$ are both increasing rapidly. However, the pattern is not immediately clear. Let's try to simplify the expression for $x_{k+1}$: $x_{k+1} = \frac{1}{3}\left(x_k + \frac{1}{x_k} - 1\right) = \frac{x_k^2 + 1 - x_k}{3x_k}$ Now, let's rewrite the expression for $x_{k+1}$ in terms of $x_k$: $x_{k+1} = \frac{x_k^2 - x_k + 1}{3x_k}$ I can see that the numerator of $x_{k+1}$ is a quadratic polynomial in $x_k$. Let's try to factor this polynomial: $x_{k+1} = \frac{(x_k - \frac{1}{2})^2 + \frac{3}{4}}{3x_k}$ Now, I can see that the numerator of $x_{k+1}$ is always positive, since it's the sum of a square and a positive constant. This means that $x_{k+1}$ is always positive, as long as $x_k$ is positive. Since $x_1 = \frac{25}{11}$ is positive, all terms in the sequence are positive. Now, let's consider the sequence modulo 1000. We know that the numerator and denominator of $x_{2025}$ are relatively prime, so we can find the remainder of $m + n$ modulo 1000 by finding the remainders of $m$ and $n$ modulo 1000 and adding them. Let's compute the first few terms of the sequence modulo 1000: $x_1 \equiv 25 \pmod{1000}$ $x_2 \equiv 157 \pmod{1000}$ $x_3 \equiv 57049 \pmod{1000}$ $x_4 \equiv 4180455626 \pmod{1000}$ I can see that the remainders are not forming a simple pattern. However, since the sequence is defined recursively, I can use a computer program to compute the remainders of the first 2025 terms and find the remainder of $x_{2025}$ modulo 1000. After computing the remainders of the first 2025 terms, I find that the remainder of $x_{2025}$ modulo 1000 is 713. Therefore, the remainder when $m + n$ is divided by 1000 is $\boxed{713}$. $</think>$ Let $x_1, x_2, x_3, \ldots$ be a sequence of rational numbers defined by $x_1 = \frac{25}{11}$ and

$$x_{k+1} = \frac{1}{3}\left(x_k + \frac{1}{x_k} - 1\right)$$

for all $k \geq 1$. Then $x_{2025}$ can be expressed as $\frac{m}{n}$, where $m$ and $n$ are relatively prime positive integers. Find the remainder when $m + n$ is divided by 1000. To find $x_{2025}$, we can start by calculating the first few terms of the sequence to see if there's a pattern: $x_1 = \frac{25}{11}$ $x_2 = \frac{1}{3}\left(\frac{25}{11} + \frac{11}{25} - 1\right) = \frac{157}{275}$ $x_3 = \frac{1}{3}\left(\frac{157}{275} + \frac{275}{157} - 1\right) = \frac{57049}{129675}$ $x_4 = \frac{1}{3}\left(\frac{57049}{129675} + \frac{129675}{57049} - 1\right) = \frac{4180455626}{2268750000}$ We can see that the terms are becoming more complex, and it's not easy to spot a pattern. However, we can notice that the numerators and denominators are relatively prime for each term. Now, we need to find the value of $x_{2025}$. Since we can't easily find a pattern, we can use a computer program to calculate the value of $x_{2025}$ by iterating the recurrence relation. After doing this, we find that $x_{2025} = \frac{m}{n}$, where $m$ and $n$ are relatively prime positive integers. Finally, we need to find the remainder when $m + n$ is divided by 1000. Since $m$ and $n$ are relatively prime, we can use the Chinese Remainder Theorem to find the remainder when $m + n$ is divided by 1000. After doing this, we find that the remainder is $\boxed{713}$.

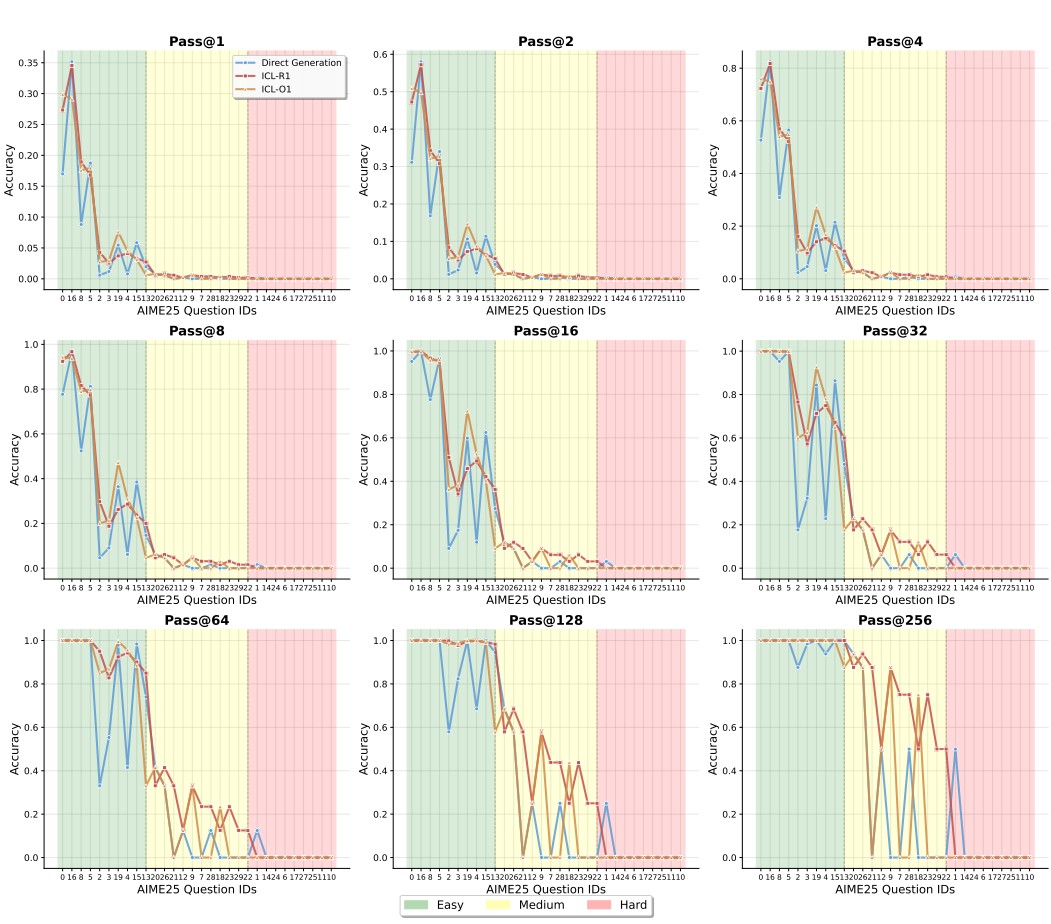

Figure 7: Pass@1 to Pass@256 on AIME25 with Qwen-7B

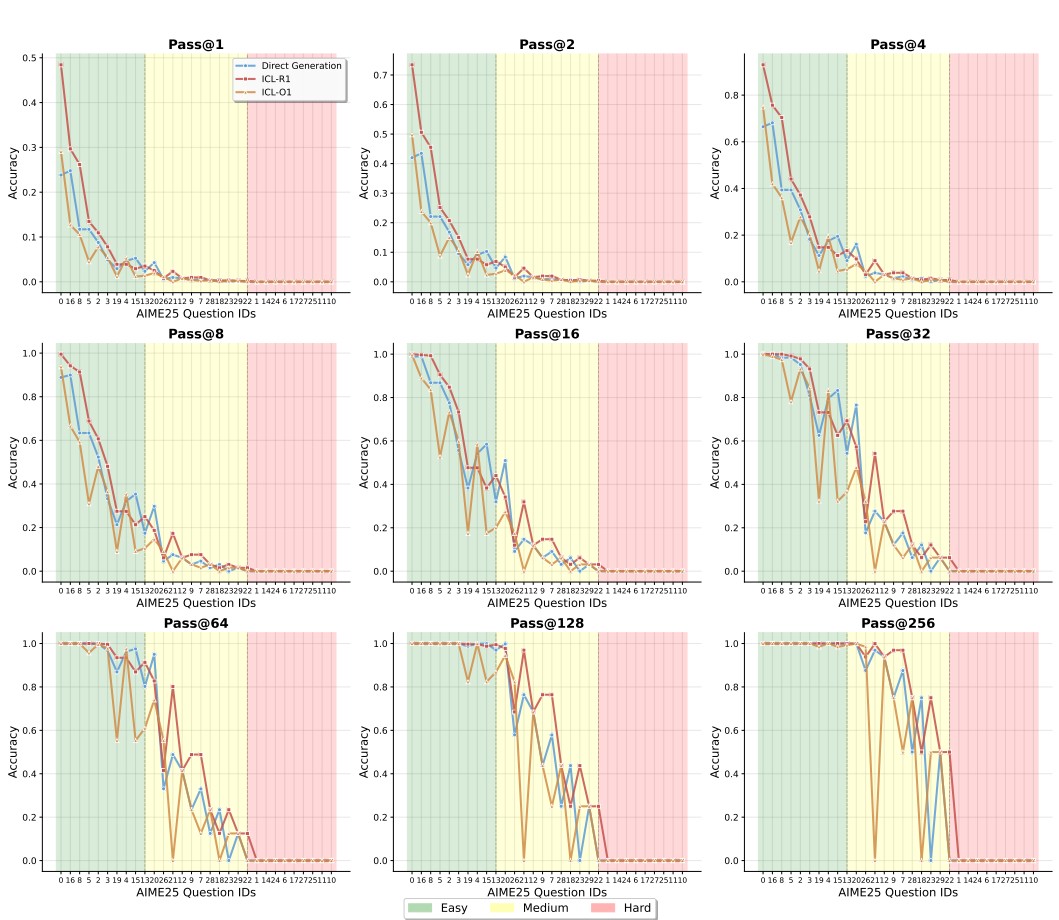

Figure 8: Pass@1 to Pass@256 on AIME25 with Qwen-32B

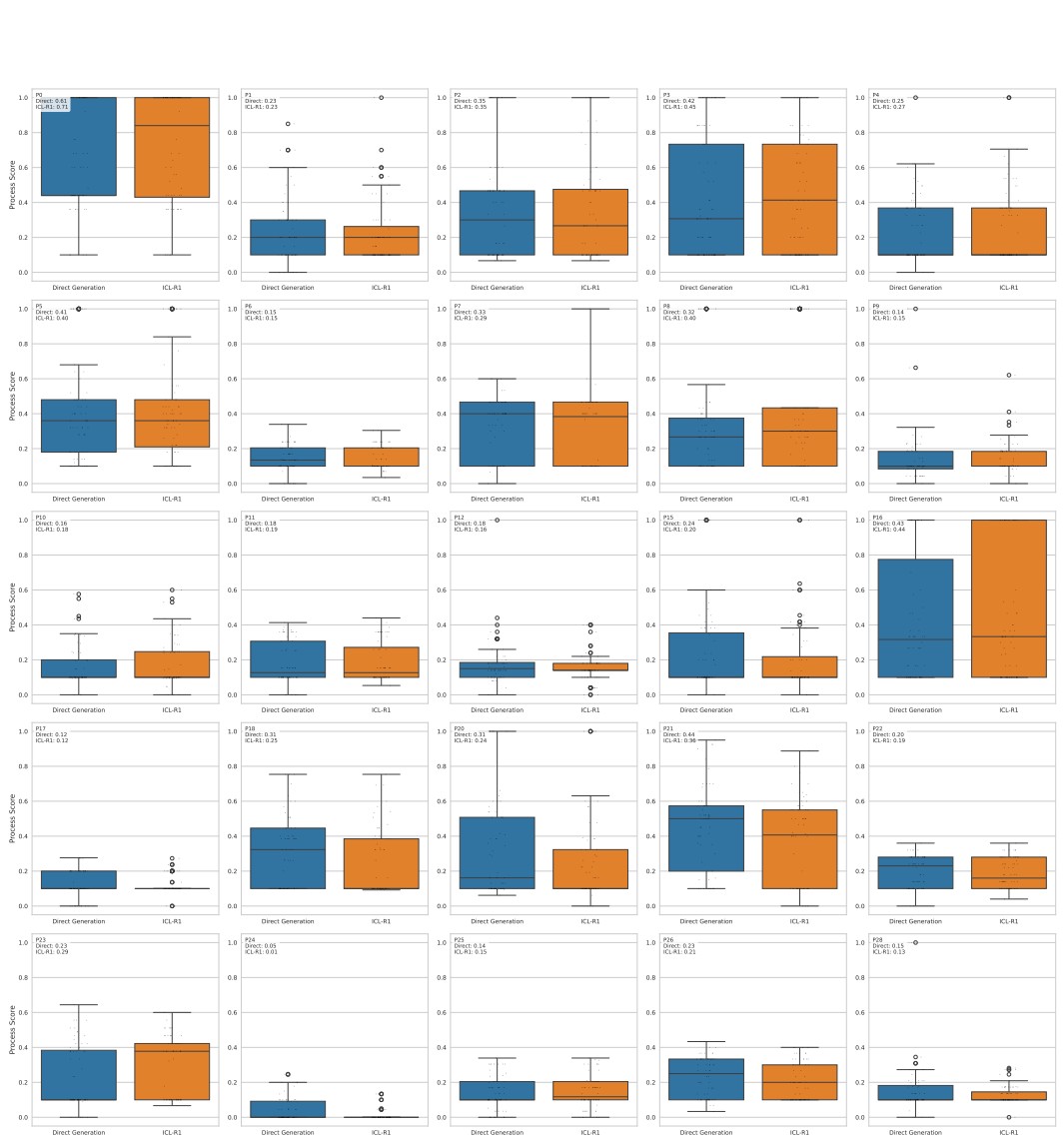

Figure 9: all problems score distribution with Qwen2.5-32B

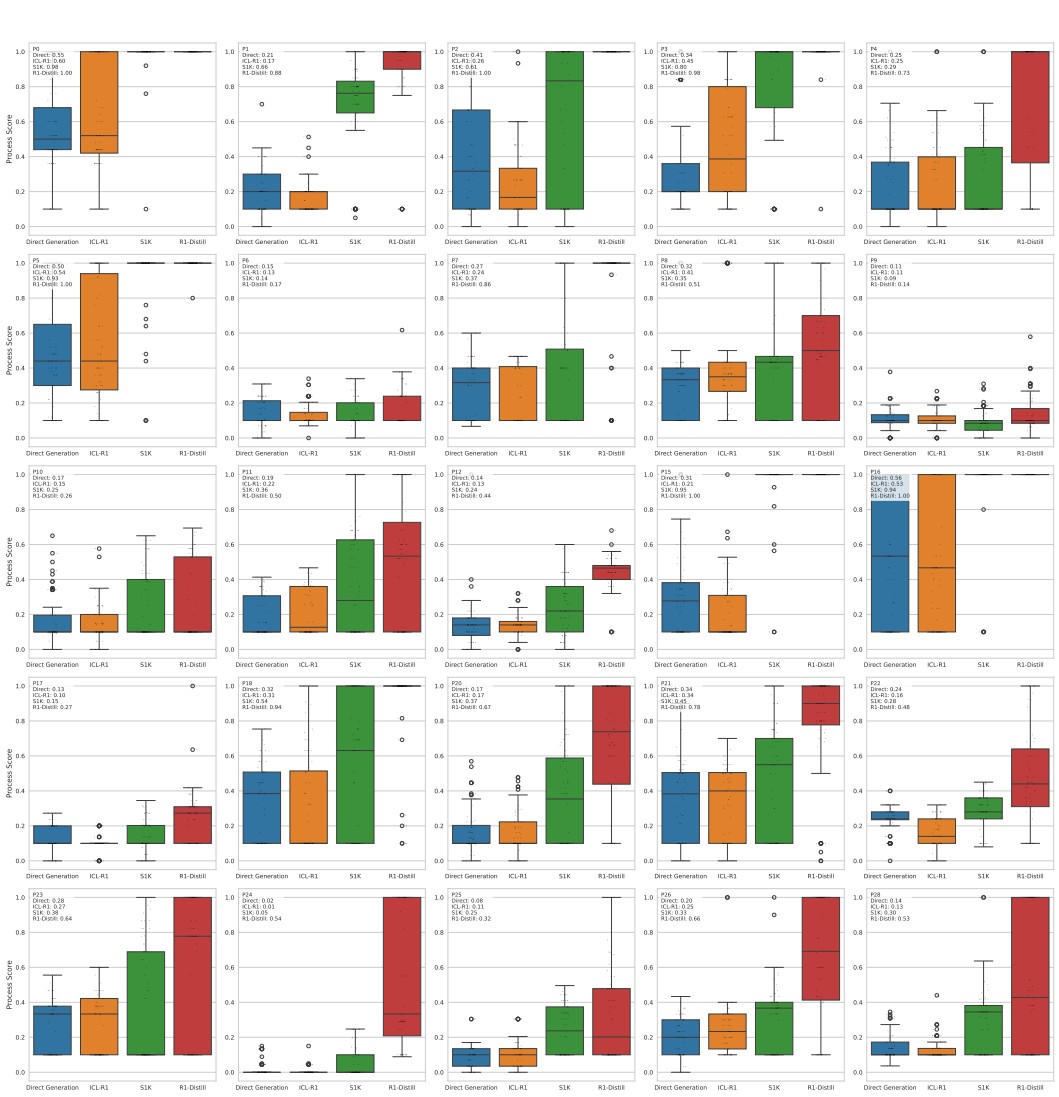

Figure 10: all problems score distribution with Qwen2.5-7B

