# OpenReview forum: "Long CoT In-Context Learning Can Empower Pre-trained LLMs"
_ICLR.cc/2026/Conference — Submitted to ICLR 2026_

### Official Review · Reviewer_D4gS · 2025-10-25

**Soundness:** 2
**Presentation:** 2
**Contribution:** 3
**Rating:** 2
**Confidence:** 5

**Summary:**

The paper Long CoT In-Context Learning Can Empower Pre-Trained LLMs investigates whether large pre-trained language models can exhibit long-CoT reasoning without any parameter updates. The authors propose in-context learning with long CoT demonstrations (Long CoT ICL) as a tuning-free mechanism to elicit reflective and verification-oriented reasoning styles. Using Qwen2.5 and DeepSeek-V3 models on mathematical benchmarks, they find that Long CoT ICL consistently induces more structured reasoning traces and improves pass@K accuracy, particularly on medium-difficulty problems. Analysis reveals that these gains stem mainly from “style activation”, the imitation of long CoT patterns rather than genuine increases in reasoning ability. When demonstrations are semantically related to the target problem, performance further improves, suggesting that Long CoT ICL can partially bridge the gap between base and fine-tuned reasoning models.

**Strengths:**

- motivation: The paper targets a crucial open question, whether reasoning capabilities can be elicited from base models without fine-tuning.

- experiments: Multiple model families and benchmarks are covered, with quantitative and qualitative analyses of reasoning depth, reflection, and correction.

- reproducibility: Appendices and algorithmic descriptions are detailed and transparent.

**Weaknesses:**

- conceptual novelty: The results largely confirm expectations that long CoT patterns can be induced via ICL but do not fundamentally improve reasoning ability.

- mathematics tasks: Claims of generalisability beyond the mathematical domain are speculative.

- Figures are too small and visually dense, making it difficult to interpret the results and follow comparisons across models and conditions; this substantially weakens readability. The authors should rethink the graphical design to highlight key takeaways.

- exposition: The paper is overly long, with several sections restating similar findings.

**Questions:**

1. How can you distinguish between genuine reasoning improvement and surface imitation of reasoning patterns beyond qualitative examples?

2- Do you expect Long CoT ICL to scale similarly in non-mathematical reasoning tasks (commonsense/multilingual/multimodal reasoning), or are the findings domain-specific?

---

### Official Review · Reviewer_F1uH · 2025-10-26

**Soundness:** 2
**Presentation:** 2
**Contribution:** 2
**Rating:** 2
**Confidence:** 3

**Summary:**

The paper focuses on supplying long chain-of-thought (CoT) demonstrations as in-context examples (ICL) which can "activate" long CoT behaviors in pre-trained LLMs, yielding accuracy gains on math benchmarks (i.e., AIME etc.), without any parameter updates. The central claims are (i) ICL with long CoT increases behaviors such as reflection/verification relative to direct generation, and (ii) this yields pass@K improvements; (iii) most of the gains come from "style activation" instead of genuine increases in "reasoning" ability.

**Strengths:**

* The paper aims to separate style activation from ability gains at test time which is an important question in helping understand LLM reasoning capabilities.

* Multiple model families and benchmarks are used along with several ablations, which is helpful to understand how robust gains/results are.

**Weaknesses:**

* Unfortunately, I am unsure about the novelty of this paper's core idea: the core idea that few-shot ICL with rationales/long traces to elicit better reasoning has been central since chain-of-thought and self-consistency first came about along with subsequent works (e.g., tree-of-thoughts, ReAct, etc.) that explicitly induce deeper reasoning patterns at inference via prompting and control strategies. This paper’s contribution is essentially: "make the in-prompt rationales longer (from an R1-style source) and you’ll see more reflective/verification behavior and small accuracy bumps." Unless I am missing something (and please point out if I am), this seems like another variant of a well-trod idea (few-shot CoT ICL), and considerably less novel than prior inference-time search/consensus methods (Self-Consistency, ToT, ReAct) that already show larger gains on math/logic tasks without finetuning. The paper does not establish a distinct technique beyond choosing longer demos and measuring a bespoke behavior ratio.

* With that said, I don't see any comparison with other strong test-time baselines (i.e., majority voting, ToT, or ReAct) which tend to be more standard/commonly-used training-free inference-time reasoning baselines that often eclipse naive CoT gains on math tasks (unless I've missed this somewhere).

* The selection of models and datasets seems a bit small in terms of these experiments: essentially, I see just three models and four datasets which makes it hard to extrapolate to other settings (e.g., other model families, other types of data/benchmarks aside math reasoning, etc.). Moreover, I don't see any confidence bands or averages across different seeds (or controlling for sequence length), which would be nice to have.

* Is there a reason why, for MATH500 and MinervaMath, only randomly selected subsets of 50 problems each are used as the test set? Additionally, results from Table 2 seem to show that the trend is non-monotonic and unstable; DeepSeek V3 degrades after 2-shot, for example.

**Questions:**

See weaknesses.

---

### Official Review · Reviewer_ihMu · 2025-10-26

**Soundness:** 3
**Presentation:** 2
**Contribution:** 3
**Rating:** 6
**Confidence:** 4

**Summary:**

This paper shows that ICL with long CoT demonstrations can elicit complex reasoning behaviors in pre-trained large language models without fine-tuning. Through experiments on mathematical reasoning benchmarks, the authors find that long CoT ICL improves accuracy and activates reflective, verification-based reasoning, revealing that base models already contain latent reasoning abilities

**Strengths:**

- The paper utilizes long CoT ICL as a parameter-free approach to probe whether pre-trained LLMs already possess latent reasoning abilities, filling a clear research gap between zero-shot prompting and fine-tuning methods.
- Experiments across multiple model families and four math reasoning benchmarks provide strong empirical evidence.
- This study conducts an in-depth analysis of long CoT ICL, hypothesizing that the improved accuracy primarily stems from style activation.

**Weaknesses:**

- The paper is limited to the math domain; it remains unclear whether the observed long CoT ICL effects generalize to other reasoning domains, such as commonsense reasoning.
- The work lacks novelty — the findings reported in this paper, such as “long CoT ICL yields greater performance gains when problem-relevant demonstrations are provided,” have already been proposed in many previous studies, such as [1] and [2].
- Presentation issues: The numbers in Figure 1 are difficult to read, and the meanings of the x- and y-axes in Figure 2 are unclear.
- Typos: “pre-trained, pre-trained” in line 479.

References

[1] Wang, Boshi, et al. "Towards understanding chain-of-thought prompting: An empirical study of what matters." arXiv preprint arXiv:2212.10001 (2022).

[2] Stechly, Kaya, Karthik Valmeekam, and Subbarao Kambhampati. "Chain of thoughtlessness? an analysis of cot in planning, 2024." URL https://arxiv. org/abs/2405.04776.

**Questions:**

Is using demonstrations generated by DeepSeek-R1 the optimal choice? Have the authors explored demonstrations produced by other models to provide a more comprehensive and fair comparison?

---

### Official Review · Reviewer_dSVS · 2025-11-01

**Soundness:** 3
**Presentation:** 3
**Contribution:** 2
**Rating:** 2
**Confidence:** 5

**Summary:**

This paper investigates whether long chain-of-thought (CoT) demonstrations can activate reasoning behavior in large language models (LLMs) without additional training. The authors use few-shot in-context learning (ICL) prompts constructed from long reasoning examples generated by a stronger model (DeepSeek-R1) and test weaker base or instruction-tuned models (Qwen2.5-7B/32B, DeepSeek-V3) on several math and logic benchmarks (AIME24/25, MATH500, MinervaMath).
They report that long-CoT demonstrations increase reasoning-related behaviors (reflection, verification, correction) and slightly improve accuracy, especially on medium-difficulty problems.
They conclude that LLMs already contain latent reasoning abilities that can be activated through prompt-based long-CoT guidance.

**Strengths:**

1. Clear experimental setup using established reasoning benchmarks.

2. Consistent and interpretable results showing how long demonstrations affect reasoning behavior.

3. Good qualitative analysis of reasoning style activation (reflection, verification, correction).

4. Solid writing and reproducibility.

**Weaknesses:**

1. The central finding — that longer CoT examples in few-shot prompts improve reasoning — is already known and widely practiced.

2. The paper revalidates existing intuition rather than introducing a new concept, method, or metric.

3. The “activation” framing is somewhat overstated; the study shows style imitation, not a deeper reasoning improvement.

4. Evaluation focuses exclusively on math-style reasoning; generalization to other domains is unclear.

5. Heavy dependence on teacher-generated CoT samples (DeepSeek-R1) limits the practical novelty.

**Questions:**

1. Can you clarify how your findings differ fundamentally from prior work on few-shot CoT prompting and test-time compute scaling?

2. Do you have evidence that the model’s reasoning improves, not merely imitates longer answers?

3. Would similar “activation” occur with random or short but well-structured examples?

4. How does this study inform future model training or prompting beyond reaffirming known behavior?

---

### Meta-Review · Area_Chair_x3pZ · 2026-01-13

**Summary:**

Reviewers agree the paper studies an interesting question—whether long chain-of-thought demonstrations in few-shot prompting can elicit more “long-CoT” behaviors without parameter updates—but they find the core contribution insufficient for acceptance. The central concern is novelty: the main takeaway largely reiterates well-known CoT/ICL behavior (“longer/more structured rationales can improve reasoning-like outputs”), and the paper does not clearly establish a distinct technical method beyond longer demonstrations and analysis metrics. Reviewers also question whether the reported gains reflect genuine reasoning improvement versus surface-level style imitation, and view the “activation” framing as overstated. Empirically, reviewers note missing comparisons to stronger, standard test-time baselines (e.g., self-consistency/majority voting, ToT, ReAct) and raise concerns about limited and potentially unstable evaluation (small benchmark/model coverage, subset sampling choices, lack of confidence intervals/seeds). Finally, generalization beyond math reasoning remains unclear, and presentation/readability issues (dense/small figures, length) further weaken the submission.

**Reviewer Concerns:**

### Concerns that appear addressed by the rebuttal
N/A: no rebuttal was given.

### Concerns that remain outstanding (based on the reviews and what is visible in the submission)
1. **Novelty / incremental contribution**: Multiple reviewers argue the main idea is a straightforward variant of standard few-shot CoT prompting (using longer rationales) and does not establish a distinct technique or insight beyond revalidating known behaviors.
2. **“Activation” vs. true reasoning improvement**: Reviewers request stronger evidence that the method improves reasoning rather than merely producing longer, more reflective-looking traces; current evidence is viewed as consistent with style imitation.
3. **Missing strong test-time baselines**: Lack of comparisons to widely used inference-time reasoning baselines (e.g., self-consistency/majority vote, ToT, ReAct) makes it hard to judge the practical and scientific significance of the gains.
4. **Experimental robustness / statistical support**: Concerns include limited model/dataset breadth, use of small random subsets for some benchmarks, non-monotonic trends across shots, and lack of confidence intervals / seed averaging / controls.
5. **Generality beyond math**: Claims about broader reasoning generalization are viewed as speculative given evaluation is concentrated on math-style benchmarks.
6. **Presentation issues**: Figures are described as dense/small and hard to read, and the paper is perceived as overly long with repetitive sections.
7. **Dependence on a specific teacher model for demonstrations**: Reviewers ask whether using demonstrations from DeepSeek-R1 is essential/optimal and whether demonstrations from other models would change conclusions.

**Reviewer Scores:**

Unchanged: no rebuttal was given.

---

### Decision · Program_Chairs · 2026-01-26

Reject